# QGFN: Controllable Greediness with Action Values

**Elaine Lau**[1][2][*]   **Stephen Zhewen Lu**[2]   **Ling Pan**[1][5]
**Doina Precup**[1][2][3]   **Emmanuel Bengio**[4]

[1]Mila - Québec AI Institute   [2]McGill University   [3]Google Deepmind
[4]Valence Labs   [5]Hong Kong University of Science and Technology

## Abstract

Generative Flow Networks (GFlowNets; GFNs) are a family of energy-based generative methods for combinatorial objects, capable of generating diverse and high-utility samples. However, consistently biasing GFNs towards producing high-utility samples is non-trivial. In this work, we leverage connections between GFNs and reinforcement learning (RL) and propose to combine the GFN policy with an action-value estimate, $Q$, to create greedier sampling policies which can be controlled by a mixing parameter. We show that several variants of the proposed method, QGFN, are able to improve on the number of high-reward samples generated in a variety of tasks without sacrificing diversity.

## 1 Introduction

Generative Flow Networks [Bengio et al., 2021a,b], also known as GFlowNets, or GFNs, were recently introduced as a novel generative framework in the family of energy-based models [Malkin et al., 2022b, Zhang et al., 2022]. Given some energy function $f(x)$ over objects $\mathcal{X}$, the promise of GFNs is to train a sampler $p_\theta$ such that at convergence $p_\theta(x) \propto \exp(-f(x))$; $\exp(-f(x))$ is also referred to as *the reward $R(x)$* in GFN literature, inheriting terminology from Reinforcement Learning (RL). GFNs achieve this sampling via a constructive approach, treating the creation of some object $x$ as a sequential additive process (rather than an iterative local process *à la* Markov chain Monte Carlo (MCMC) that can suffer from mode-mixing issues). The main advantage of a GFN is its ability to generate a greater diversity of low-energy/high-reward objects compared to approaches based on MCMC or RL [Bengio et al., 2021a, Jain et al., 2022, 2023], or even Soft-RL–which, while related to GFNs, accomplishes something different by default [Tiapkin et al., 2023, Mohammadpour et al., 2024, Deleu et al., 2024].

To generate more interesting samples and avoid oversampling from low-reward regions, it is common to train a model to sample in proportion to $R(x)^\beta$; $\beta$ is an inverse temperature, typically $\gg 1$, which pushes the model to become *greedier*. The use of this temperature (hyper)parameter is an important control knob in GFNs. However, tuning this hyperparameter is non-trivial, which complicates training certain GFNs; for example, trajectory balance [Malkin et al., 2022a] is sensitive to the "peakiness" of the reward landscape [Madan et al., 2023]. Although it is possible to train temperature-conditional models [Kim et al., 2023a], doing so essentially requires learning a whole *family* of GFNs–no easy task, albeit doable, e.g., in multiobjective settings [Jain et al., 2023].

In this work, we propose an approach that allows selecting arbitrary greediness at inference time, which preserves the generality of temperature-conditionals, while simplifying the training process. We do so without the cost of learning a complicated family of functions and without conditionals, instead only training two models: a GFlowNet and an action-value function $Q$ [Watkins and Dayan, 1992, Mnih et al., 2013].

38th Conference on Neural Information Processing Systems (NeurIPS 2024).
Correspondence to: Elaine Lau <tsoi.lau@mail.mcgill.ca>
Source code available at: https://github.com/yunglau/QGFN/
[*]This work was done during an internship at Valence Labs.

Armed with the forward policy of a GFN, $P_F$ (which decides a probability distribution over actions given the current state, i.e. $\pi$ in RL), and the action-value, $Q$, we show that it is possible to create a variety of controllably greedy sampling policies, controlled by parameters that require no retraining. We show that it is possible to simultaneously learn $P_F$ and $Q$, and in doing so, to generate more high-reward yet diverse object sets. In particular, we introduce and benchmark three specific variants of our approach, which we call QGFN: $p$-greedy, $p$-quantile, and $p$-of-max.

We evaluate the proposed methods on 5 standard tasks used in prior GFN works: the fragment-based molecular design task introduced by Bengio et al. [2021a], 2 RNA design tasks introduced by Sinai et al. [2020], a small molecule design task based on QM9 [Jain et al., 2023], as well as a bit sequence task from Malkin et al. [2022a], Shen et al. [2023]. The proposed method outperforms strong baselines, achieving high average rewards and discovering modes more efficiently, sometimes by a large margin. We conduct an analysis of the proposed methods, investigate key design choices, and probe the methods to understand why they work. We also investigate other possible combinations of $Q$ and $P_F$–again, entirely possible at inference time, by combining trained $Q$ and $P_F$ models.

## 2   Background and Related Work

We follow the general setting of previous GFN literature and consider the generation of discrete finite objects, but in principle our method could be extended to the continuous case [Lahlou et al., 2023].

**GFlowNets**   GFNs [Bengio et al., 2021b] sample objects by decomposing their generation process in a sequence $\tau = (s_0, .., s_T)$ of constructive steps. The space can be described by a pointed directed acyclic graph (DAG) $\mathcal{G} = (\mathcal{S}, \mathcal{A})$, where $s \in \mathcal{S}$ is a partially constructed object, and $(s \rightarrow s') \in \mathcal{A} \subset \mathcal{S} \times \mathcal{S}$ is a valid additive step (e.g., adding a fragment to a molecule). $\mathcal{G}$ is rooted at a unique initial state $s_0$.

GFNs are trained by pushing a model to satisfy so-called *balance* conditions of flow, whereby flows $F(s)$ going through states are conserved such that terminal states (corresponding to fully constructed objects) are sinks that absorb $R(s)$ (non-negative) units of flow, and intermediate states have as much flow coming into them (from parents) as flow coming out of them (to children). This can be described succinctly as follows, for any partial trajectory $(s_n, .., s_m)$:

$$F(s_n) \prod_{i=n}^{m-1} P_F(s_{i+1}|s_i) = F(s_m) \prod_{i=n}^{m-1} P_B(s_i|s_{i+1}) \tag{1}$$

where $P_F$ and $P_B$, the forward and backward policies, are distributions over children and parents respectively, representing the fraction of flow emanating forward and backward from a state. By construction for terminal (leaf) states $F(s) = R(s)$.

Balance conditions lead to learning objectives such as Trajectory Balance [TB; Malkin et al., 2022a], where $n = 0$ and $m$ is the trajectory length, and Sub-trajectory Balance [SubTB; Madan et al., 2023], where all combinations of $(n, m)$ are used. While a variety of GFN objectives exist, we use these two as they are considered standard. If those objectives are fully satisfied, i.e. 0-loss everywhere, terminal states are guaranteed to be sampled with probability $\propto R(s)$ [Bengio et al., 2021a].

**Action values**   For a broad overview of RL, see Sutton and Barto [2018]. A central object in RL is the *action-value* function $Q^\pi(s, a)$, which estimates the expected "reward-to-go" when following a policy $\pi$ starting in some state $s$ and taking action $a$; for some discount $0 \leq \gamma \leq 1$,

$$Q^\pi(s, a) = \mathbb{E}_{\substack{a_t \sim \pi(\cdot|s_t) \\ s_{t+1} \sim T(s_t, a_t)}} \left[ \sum_{t=0}^{\infty} \gamma^t R(s_t) \big| s_0 = s, a_0 = a \right] \tag{2}$$

While $T(s, a)$ can be a stochastic transition operator, in a GFN context objects are constructed in a deterministic way (although there are stochastic GFN extensions; Pan et al. [2023b]). Because rewards are only available for finished objects, $R(s) = 0$ unless $s$ is a terminal state, and we use $\gamma = 1$ to avoid arbitrarily penalizing "larger" objects. Finally, as there are several possible choices for $\pi$, we will simply refer to $Q^\pi$ as $Q$ when statements apply to a large number of such choices.

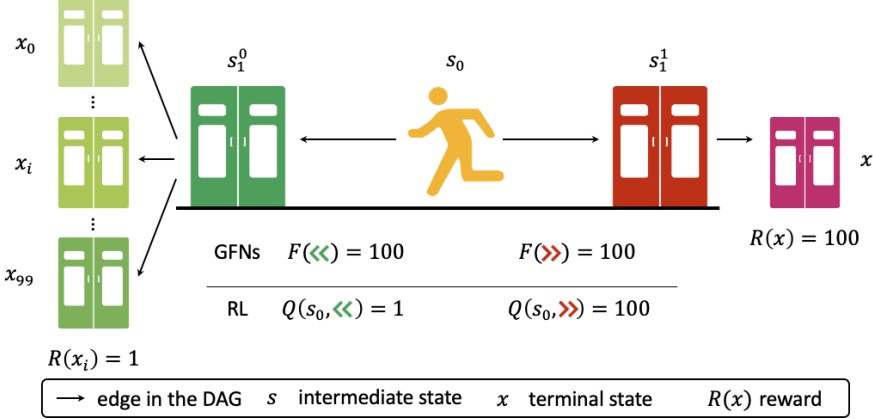

Figure 1: Solely relying on flow functions $F$ in GFNs can be insufficient. While GFNs capture *how much stuff* there is, they spend time sampling from lots of small rewards.

## 2.1 Related Work

**RL and GFlowNets**  There are clear connections between the GFN framework and RL framework [Tiapkin et al., 2023, Mohammadpour et al., 2024, Deleu et al., 2024]. Notably, Tiapkin et al. [2023] show that it is possible to reformulate fixed-$P_B$ GFlowNets as a Soft-RL problem within a specific class of reward-modified MDPs. While they show that this reformulated problem can then be tackled with any Soft-RL method, this still essentially solves the original GFlowNet problem, i.e. learn $p_\theta(x) \propto R(x)$. Instead, we are interested in greedier-yet-diverse methods.

A commonly used tool in GFNs (and QGFN) to increase the average reward, aka "greediness" of the drawn samples, is to adapt the reward distribution by using an altered reward function $\hat{R}(x) = R(x)^\beta$ and adjusting the exponent parameter $\beta$: the higher the $\beta$, the greedier the model should be [Jain et al., 2023]. However, increasing $\beta$ often induces greater numerical instability (even on a log scale), and reduces diversity because the model is less incentivized to explore "middle-reward" regions. This can lead to mode collapse. Kim et al. [2023a] show that it is possible to train models that are conditioned on $\beta$, which somewhat alleviates these issues, but at the cost of training a more complex model.

Again, while we could leverage the equivalence between the GFN framework and the Soft-RL framework [Tiapkin et al., 2023], this approach would produce a soft policy. We propose a different approach that increases greediness of the policy via "Hard-RL".

**Improving GFlowNet sampling**  A number of works have also made contributions towards improving utility in GFlowNets, via local search [Kim et al., 2023b], utilizing intermediate signals [Pan et al., 2023a], or favoring high-potential-utility intermediate states [Shen et al., 2023], as well as the use of RL tools such as replay buffers [Vemgal et al., 2023], target networks [Lau et al., 2023], or Thompson sampling [Rector-Brooks et al., 2023].

## 3 Motivation

Consider the following scenario, illustrated in Figure 1: an agent is faced with two doors. Behind the left door, there are 100 other doors, each hiding a reward of 1. Behind the right door, there is a single door hiding a reward of 100. The flow will be such that $F(\text{left}) = F(\text{right}) = 100$, meaning that a GFN agent will pick either door with probability $1/2$. The action value function is $Q(s_0, \text{left}) = 1$, $Q(s_0, \text{right}) = 100$, so an agent basing its decisions on $Q$ will reach for the door with reward 100.

This example shows that *relying solely on flows is not always sufficient to provide high-value samples frequently* and is illustrative of real-world scenarios. Consider molecular design (a very large search space) with some reward in $[0, 1]$; there may be $10^6$ molecules with reward .9, but just a dozen with reward 1. Since $.9 \times 10^6$ is much bigger than $12 \times 1$, the probability of sampling a reward 1 molecule will be low if one uses this reward naively. While using a temperature parameter is a useful way to increase the probability of the reward 1 molecules, we propose a complementary, inference-time-adjustable method.

Note that relying solely on $Q$ is also insufficient. If $Q$ were estimated very well for the optimal policy (which is extremely hard), it would be (somewhat) easy to find the reward 1 molecules via some kind of tree search following $Q$ values. However, in practice, RL algorithms easily collapse to non-diverse solutions, only discovering a few high reward outcomes. This is where flows are useful: because they capture *how much stuff* there is in a particular branch (rather than an expectation), it is useful to follow flows to find regions where there is potential for reward. In this paper, we propose a method that can be greedier (by following $Q$) while still being exploratory and diverse (by following $F$ through $P_F$).

## 4 QGFN: controllable greediness through $Q$

Leveraging the intuition from the example above, we now propose and investigate several ways in which we can use $Q$-functions to achieve our goal; we call this general idea **QGFN**. In particular, we present three variants of this idea, which are easy to implement and effective: $p$-greedy QGFNs, $p$-quantile QGFNs, and $p$-of-max QGFNs. In §5 and §6, we show that these approaches provide a favourable trade-off between reward and diversity, during both training and inference.

As is common in GFlowNets, we train QGFN by sampling data from some behavior policy $\mu$. We train $F$ and $P_F$ (and use a uniform $P_B$) to minimize a flow balance loss on the minibatch of sampled data, using a temperature parameter $\beta$. Additionally, we train a $Q$-network to predict action values on the same minibatch (the choice of loss will be detailed later). Training the GFN and $Q$ on a variety of behaviors $\mu$ is possible because both are off-policy methods. Indeed, instead of choosing $\mu$ to be a noisy version of $P_F$ as is usual for GFNs, we combine the predictions of $P_F$ and $Q$ to form a *greedier* behavior policy. In all proposed variants, this combination is modulated by a factor $p \in [0, 1]$, where $p = 0$ means that $\mu$ depends only on $P_F$, and $p = 1$ means $\mu$ is greediest, as reflected by $Q$. The variants differ in the details of this combination.

$p$**-greedy QGFN** Here, we define $\mu$ as a mixture between $P_F$ and the $Q$-greedy policy, controlled by factor $p$:

$$\mu(s'|s) = (1 - p)P_F(s'|s) + p\mathbb{I}[s' = \mathrm{argmax}_i Q(s, i)] \tag{3}$$

In other words, we follow $P_F$, but with probability $p$, the greedy action according to $Q$ is picked. All states reachable by $P_F$ are still reachable by $\mu$. Note that $p$ can be changed to produce very different $\mu$ without having to retrain anything.

$p$**-quantile QGFN** Here, we define $\mu$ as a masked version of $P_F$, where actions below the $p$-quantile of $Q$, denoted $q_p(Q, s)$, have probability 0 (so are discarded):

$$\mu(s'|s) \propto P_F(s'|s)\mathbb{I}[Q(s, s') \geq q_p(Q, s)] \tag{4}$$

This can be implemented by sorting $Q$ and masking the logits of $P_F$ accordingly. This method is more aggressive, since it prunes the search space, potentially making some states unreachable. Again, $p$ is changeable.

$p$**-of-max QGFN** Here, we define $\mu$ as a masked version of $P_F$, where actions with $Q$-values less than $p \max_a Q(s, a)$ have probability 0:

$$\mu(s'|s) \propto P_F(s'|s)\mathbb{I}[Q(s, s') \geq p \max_i Q(s, i)] \tag{5}$$

This is similar to $p$-quantile pruning, but the number of pruned actions changes as a function of $Q$. If all actions are estimated as good enough, it may be that no action is pruned, and vice versa, only the best action may be retained is none of the others are good. This method also prunes the search space, and $p$ remains changeable. Note that in a GFN context, rewards are strictly positive, so $Q$ is also positive.

**Policy evaluation, or optimal control?** In the design of the proposed method, we were faced with an interesting choice: what policy $\pi$ should $Q^\pi$ evaluate? The first obvious choice is to perform $Q$-learning [Mnih et al., 2013], and estimate the optimal value function $Q^*$, with a 1-step TD objective. As we detail in §6, this proved to be fairly hard, and 1-step $Q_\theta$ ended up being a poor approximation.

A commonly used trick to improve the performance of bootstrapping algorithms is to use $n$-step returns [Hessel et al., 2018]. This proved essential to our work, and also revealed something curious:

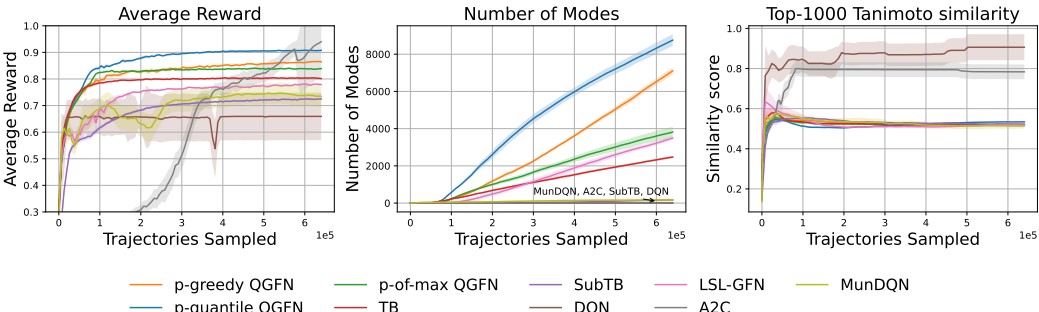

Figure 2: Fragment-based molecule task. *Left:* Average rewards over the training trajectories. *Center:* Number of unique modes with a reward threshold exceeding 0.97 and pairwise Tanimoto similarity score less than 0.65. *Right:* Average pairwise Tanimoto similarity score for the top 1000 molecules sampled by reward. Lines are the interquartile mean and standard error calculated over 5 seeds.

we've consistently found that, while results started improving at $n \geq 5$, a consistently good value of $n$ was close to the maximum trajectory length. This has an interesting interpretation, as beyond a certain value of $n$, $Q_\theta$ becomes closer to $Q^\mu$ and further from $Q^*$. In other words, using an "on-policy" estimate $Q^\mu$ rather than an estimate of the optimal policy seems beneficial, or at least easier to learn as a whole. In hindsight, this makes sense because on-policy evaluation is easier than learning $Q^*$, and since we are combining the $Q$-values with $P_F$, any method which biases $\mu$ correctly towards better actions is sufficient (we do not need to know exactly the best action, or its exact value).

**Selecting greediness**   In the methods proposed above, $p$ can be changed arbitrarily. We first note that we train with a constant or annealed[2] value of $p$ and treat it as a standard hyperparameter in all the results reported in §5.

Second, as discussed in §6, after training, $p$ can be changed with a predictable effect: the closer $p$ is to 1, the greedier $\mu$ becomes. Presumably, this is because the model generalizes, and $Q$-value estimates for "off-policy" actions are still a reliable guess of the reward obtainable down some particular branch. When making $p$ higher, $Q$ may remain a good *lower bound* of the expected reward (after all, $\mu$ is becoming greedier), which is still helpful. Generally, such a policy will have reasonable outcomes, regardless of the specific $\mu$ and $p$ used during training. Finally, it may be possible and desirable to use more complex schedules for $p$, or to sample $p$ during training from some (adaptive) distribution, but we leave this for future work.

# 5   Main results

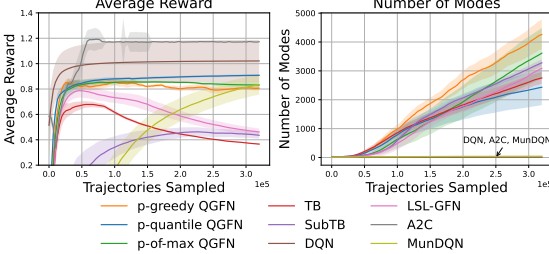

Figure 3: QM9 task. *Left:* Average rewards over training trajectories. *Right:* Number of modes with a reward above 1.10 and pairwise Tanimoto similarity less than 0.70.

We experiment on 5 standard tasks used in prior GFlowNet literature. As baselines, we use Trajectory Balance, Sub-Trajectory Balance, LSL-GFN [Kim et al., 2023a] i.e. *learning to scale logits* which controls greediness through temperature-conditioning, and as RL baselines A2C [Mnih et al., 2016], DQN [Mnih et al., 2013] (which on its own systematically underperforms in these tasks), and Tiapkin et al. [2023]'s MunDQN/GFlowNet.

We report the average reward obtained by the agents, as well as the total number of modes of the distribution of interest found during training. By *mode*, we mean a high-reward object that is separated from previously found modes by some distance threshold. The distance function and

---

[2]Specifically for $p$-quantile QGFN and $p$-of-max, we found training to be more stable if we started with $p = 0$ and annealed $p$ towards its final value with a single ½-period cosine schedule over 1500 steps.

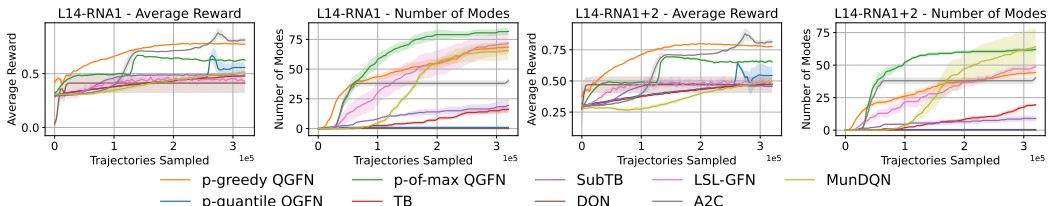

Figure 4: RNA-binding tasks, Average reward and modes. *Left*: L14RNA1 task. *Right*: L14RNA1+2 task, based on 5 seeds (interquartile mean and standard error shown).

threshold we use, as well as the minimum reward threshold for an object to be considered a mode, depend on the task.

**Fragment-based molecule generation task:**[3] Generate a graph of up to 9 fragments, where the reward is based on a prediction of the binding affinity to the sEH protein, using a model provided by Bengio et al. [2021a]. $|\mathcal{X}| \approx 10^{100}$, there are 72 available fragments, some with many possible attachment points. We use Tanimoto similarity [Bender and Glen, 2004], with a threshold of 0.65, and a reward threshold of 0.97. Results are shown in Fig. 2.

**Atom Based QM9 task:** Generate small molecules of up to 9 atoms following the QM9 dataset [Ramakrishnan et al., 2014]. $|\mathcal{X}| \approx 10^{12}$, the action space includes adding atoms or bonds, setting node or bond properties and stop. A MXMNet proxy model [Zhang et al., 2020], trained on QM9, predicts the HOMO-LUMO gap, a key indicator of molecular properties including stability and reactivity, and is used as the reward. Rewards are in the $[0, 2]$ range, with a $1.10$ threshold and a minimum Tanimoto similarity of 0.70 to define modes. Results are shown in Fig. 3.

**RNA-binding task:** Generate a string of 14 nucleobases. The reward is a predicted binding affinity to the target transcription factor, provided by the ViennaRNA [Lorenz et al., 2011] package for the binding landscapes; we experiment with two RNA binding tasks; L14-RNA1, and L14-RNA1+2 (two binding targets) with optima computed from Sinai et al. [2020]. $|\mathcal{X}|$ is $4^{14} \approx 10^{9}$, there are 4 tokens: adenine (A), cytosine (C), guanine (G), uracil (U). Results are shown in Fig. 4.

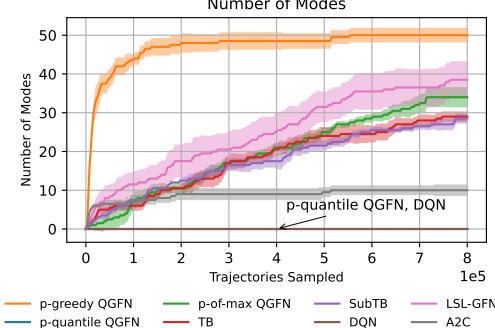

Figure 5: Bit sequence task, $k = 1$. Interquartile mean and standard error over 5 seeds.

**Prepend-Append bit sequences:** Generate a bit sequence of length 120 in a prepend-append MDP, where $|\mathcal{X}|$, limited to $\{0, 1\}^n$, is $2^{120} \approx 10^{36}$. For a sequence of length $n$, $R(x) = \exp\left(1 - \min_{y \in M} d(x, y)/n\right)$. A sequence is considered a mode if it is within edit distance $\delta$ from $M$, where $M$ is defined as per Malkin et al. [2022a] (although the task we consider here is a more complex version, introduced by Shen et al. [2023], since prepend actions induce a DAG instead of a simpler tree). In our experiment, $|M| = 60, n = 120, k = 1, \delta = 28$, where $k$ is the bit width of actions. Results are shown in Fig. 5.

## 5.1 Analysis of results

Across tasks, QGFN variants produce high rewards *and* find a higher number of modes, i.e. high-reward dissimilar objects. The latter could seem surprising, because a priori, increasing the greediness of a method likely reduces its diversity. This fundamental trade-off is known in RL as the exploration-exploitation dilemma [Sutton, 1988, Sutton and Barto, 2018]. However, we are leveraging two methods and combining their strengths to reap the best from both worlds: GFlowNets are able to *cover* the state space, because they attempt to model all of it, by learning $p_\theta(x) \propto R(x)$, while $Q$ approximates the expected reward of a particular action, which can help guide the agent by selecting

---

[3]We note that there exist a number of differing implementations of this task in the GFlowNet literature, we use that of https://github.com/recursionpharma/gflownet

Table 1: Fragment-based molecule task: Reward and Diversity at inference after training.

| METHOD | REWARD($\uparrow$) | SIMILARITY ($\downarrow$) |
|---|---|---|
| GFN-TB | 0.780±0.003 | 0.545±0.002 |
| GFN-SUBTB | 0.716±0.006 | 0.513±0.003 |
| LSL-GFN | 0.717±0.020 | 0.689±0.062 |
| P-GREEDY QGFN | 0.950±0.004 | 0.551±0.015 |
| P-OF-MAX QGFN | **0.969±0.003** | 0.514±0.001 |
| P-QUANTILE QGFN | 0.955±0.003 | **0.509±0.008** |

high-expected-reward branches. Another way to think about this: GFNs are able to estimate *how many* high-reward objects there are in different parts of the state space. The agent thus ends up going in all important regions of the state space, but by being a bit more greedy through $Q$, it focuses on higher reward objects, so it is more likely to find objects with reward past the mode threshold. To further understand the performance of QGFN, we formally analyse a bandit setting, and include derivations to illustrate the general case, in Appendix §A.

We also report the average reward and pairwise similarity for the fragment task based on 1000 samples over 5 seeds taken after training in Table 1. Again, QGFNs outperform GFNs in reward, while retaining low levels of inter-sample similarity. We again note that at inference, we are able to use a different (and better) $p$ value than the one used at training time. We expand on this in §6, and show that it is easy to tune $p$ to achieve different reward-diversity trade-offs at inference. The exact $p$ values used for Table 1 are provided in Appendix §E. Also note that in LSL-GFN $\beta$ is tunable at inference, and in Table 1 we choose the $\beta$ value such that average similarity is near the 0.65 mode threshold we use (choosing a greedier $\beta$ induces a collapse in diversity).

**QGFN variants matter** We point the reader to an interesting result, which is consistent with our understanding of the proposed method. In the fragment task, the number of actions available to the agent is quite large, ranging from about 100 to 1000 actions depending on the state, and the best performing QGFN variant is one that consistently masks most actions: $p$-quantile QGFN. It is likely indeed that most actions are harmful, as combining two fragments that do not go together may be irreversibly bad, and masking helps the agent avoid undesirable regions of the state space. However, masking a fixed ratio of actions can provide more stable training.

On the other hand, in the RNA design task, there are only 5 actions (4 nucleotides ACGU & stop). We find that masking a constant *number* of actions is harmful–it is likely that in some states all of them are relevant. So, in that task, $p$-greedy and $p$-of-max QGFN work best. This is also the case in the bit sequence task, for the same reasons (see Fig. 5). To confirm this, we repeat the bit sequence task but with an expanded action space consisting not just of $\{0, 1\}$, but of all 16 ($2^4$) sequences of 4 bits, i.e. $\{0000, 0001, .., 1111\}$. We find, as shown in Fig 18, that $p$-quantile indeed no longer underperforms.

## 6 Method Analysis

We now analyze the key design choices in QGFN. We start by investigating the impact of $n$ (the number of bootstrapping steps in $Q$-Learning) and $p$ (the mixture parameter) on *training*. We then look at trained models, and reuse the learned $Q$ and $P_F$ to show that it is possible to use a variety of sampling strategies, and to change the mixture factor $p$ to obtain a spectrum of greediness at test time. Finally, we empirically probe models to provide evidence as to *why* QGFN is helpful.

**Impact of $\beta$ in QGFN** Fig. 6 shows the effect of training with different $\beta$ values on the average reward and number of modes when taking 1000 samples after training is done in the fragment task (over 5 seeds). As predicted, increasing $\beta$ increases the average reward of the agent, but at some point, causes it to become dramatically less diverse. As discussed earlier, this is typical of GFNs with a too high $\beta$, and is caused by a collapse around high-reward points and an inability for the model to further explore. While QGFN is also affected by this, it does not require as drastic values of $\beta$ to obtain a high average reward and discover a high number of modes.

**Impact of $n$ in QGFN** As mentioned in §4, training $Q$ with 1-step returns is ineffective and produces less useful approximations of the action value. Fig. 6 shows the number of modes within 1000 post-training samples in the fragment tasks, for models trained with a variety of $n$-step values.

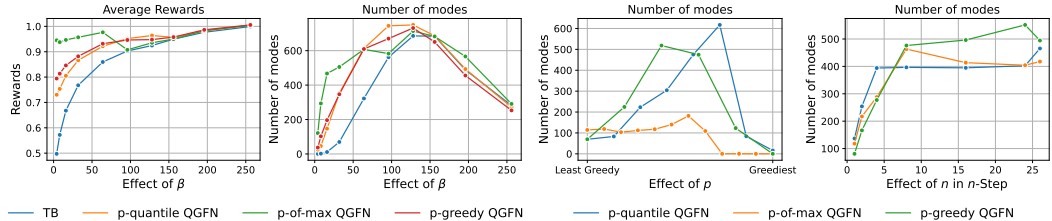

Figure 6: Fragment task. *Left*: Effect of $\beta$: Increasing greediness through $\beta$ increases the average reward but may lead to diversity collapse. QGFN maintains diversity with a lower $\beta$, while GFN collapses. Modes are counted from 1000 samples at inference, using an inference-adjusted $p$. *Right*: Effect of training parameters $p$, and $n$: Changing $p$ can control greediness, while increasing $n$ is generally beneficial. Modes are counted from 1000 samples generated using the training $p$.

Models start being consistently good at $n = 5$ and values close to the maximum length of a trajectory tend to work well too.

**Impact of the training $p$ in QGFN** While our method allows changing $p$ more or less at will, we still require some value during training. Fig. 6 shows that there are clear trade-offs between choices of $p$, some yielding significantly better diversity than others. For example, $p$-of-max is fairly sensitive to the chosen value during training, and for the fragment task doesn't seem to perform particularly well during training (especially when not annealed). On the other hand, as we will see in the next paragraph (and is also seen in Fig. 6), $p$-of-max excels at inference, and is able to generate diverse and high-reward samples by adjusting $p$.

**Changing strategy after training** We now look at the impact of changing the mixture parameter $p$ and the sampling strategy for already trained models on average reward and average pairwise similarity. We use the parameters of a model trained with $p$-greedy QGFN, $p = 0.4$.

With this model, we sample 512 new trajectories for a series of different $p$ values. For $p$-greedy and $p$-quantile, we vary $p$ between 0 and 1; for $p$-of-max, we vary $p$ between .9 and 1 (values below .9 have minor effects). We visualize the effect of $p$ on reward and similarity statistics in Fig. 9.

First, we note that increasing $p$ has the effect we would hope, increasing the average reward. Second, we note that this works without any retraining; even though we (a) use values of $p$ different than those used during training, and (b) use QGFN variants different than those used during training, the behavior is consistent: $p$ controls greediness. Let us emphasize (b): even though we trained this $Q$ with $p$-greedy QGFN, we are able to use the $Q(s, a)$ predictions just fine with entirely different sampling strategies. This has some interesting implications; most importantly, it can be undesirable to *train* with too high values of $p$ (because it may reduce the diversity to which the model is exposed), but what is learned transfers well to sampling new, high-reward objects with different values of $p$ and sampling strategies.

Finally, these results suggest that we should be able to prototype new QGFN variants, including expensive ones (e.g. MCTS) without having to retrain anything. We illustrate the performance of a few other variants in §B.2, Fig. 12.

**Is $Q$ calibrated?** For our intuition of why QGFN works to really pan out, $Q$ has to be accurate enough to provide useful guidance towards high-reward objects. We verify that this is the case with the following experiment. We take a trained QGFN model ($p$-greedy, $p = 0.4$, fragment task, maximum $n$-step) and sample 64 trajectories. For each of those trajectories, we take a random state within the trajectory as a starting point, thereafter generating 512 new tra-

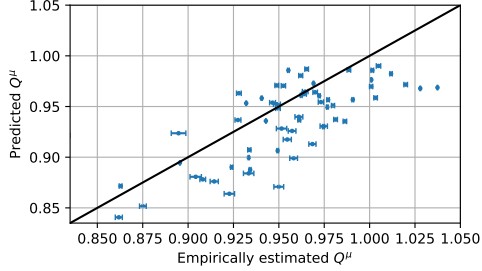

Figure 7: Comparing $Q(s, a; \theta)$ predictions with empirical estimates obtained by rollouts. Bars are standard error. $Q$ is relatively capable to estimate the returns of the corresponding policy.

jectories. We then use the reward of those 512 trajectories as an empirical estimate $\hat{Q}$ of the expected

return, which $Q$ should roughly be predicting. Fig. 7 shows that this is indeed the case. Although $Q$ is not perfect, and appears to be underestimating $\hat{Q}$, it is making reasonable predictions.

**Is $Q$ really helpful?** In this experiment, we verify our intuition that pruning based on $Q$-values is helpful. We again take a trained model for the fragment task, and sample 512 trajectories. We use $p$-of-max QGFN ($p = 0.95$), and compare it to a strategy estimating *best pruned actions*: for each trajectory, after some random number of steps $t \sim U[4, 20]$ (the max is 27), we start deterministically selecting the action that is the most probable according to $P_F$ but would be masked according to $Q$. To ensure that this is a valid thing to do, we also simply look at *Best actions*, i.e. after $t \sim U[4, 20]$ steps, deterministically select the action that is the most probable according to $P_F$, regardless of $Q$.

Fig. 8 shows that our sanity check, *Best actions*, receives reasonable rewards, while selecting actions that would have been pruned leads to much lower rewards. The average likelihood from $P_F$ of these pruned actions was .035, while the average number of total actions was $\approx 382$ (and $1/382 \approx 0.0026$). This confirms our hypothesis that $Q$ indeed masks actions that are likely according to $P_F$ but that do **not** consistently lead to high rewards.

**Why does changing $p$ work?** Recall that for QGFN to be successful, we rely on $n$-step TD, and therefore on somewhat "on-policy" estimates of $Q^\mu$. $\mu$ is really $\mu_p$, a function of $p$, meaning that if we change $p$, say to $p'$, during inference, $Q^{\mu_p}$ is not an accurate estimate of $Q^{\mu_{p'}}$. If this is the case, then there must be a reason why it is still helpful to prune based on $Q^{\mu_p}$ while using $\mu_{p'}$. In Fig. 10, we perform the

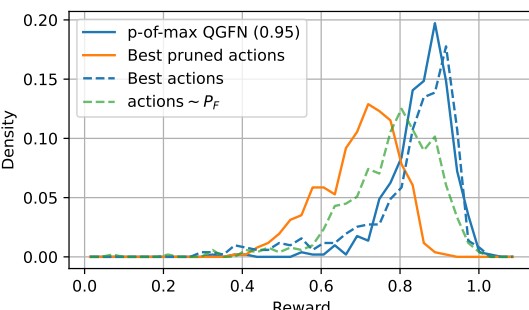

Figure 8: Pruning helps avoid low-reward parts of the state space. Reward distributions when (a) sampling with $p$-of-max; (b) greedily according to $P_F$ selecting actions that $p$-of-max would prune, *Best pruned actions*; (c) selecting most likely $P_F$ actions regardless of $Q$, *Best actions*; and (d) normal sampling from $P_F$ (without using $Q$).

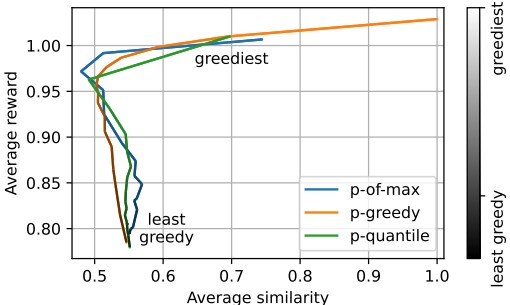

Figure 9: Varying $p$ at inference time induces reward-diversity trade-offs; fragment task.

same measurement as in Fig. 7, but we change the $p$ value used to measure $\hat{Q}^{\mu_{p'}}$. We find that, while the rank correlation drastically goes down (although it stays well above 0), $Q^{\mu_p}$ remains helpful because it *lower bounds* $\hat{Q}^{\mu_{p'}}$. If we prune based on $Q^{\mu_p}$, then we would want it to not lead us astray, and *at least* make us greedier as we increase $p$. This means that if an action is not pruned, then we expect samples coming from it to be *at least as good* as what $Q^{\mu_p}(s, a)$ predicts (in expectation). This is indeed the case.

Note that reducing $p$ simply leads $\mu$ to behave more like $P_F$, which is still a good sampler, and to rely less on $Q$, whose imperfections will then have less effect anyways.

## 7 Conclusion and Discussion

In this paper, we showed that by jointly training GFNs and $Q$-functions, we can combine their predictions to form behavior policies that are able to sample larger numbers of diverse and high-reward objects. These policies' mixture parameter $p$ is adjustable, even after training, to modulate the greediness of the resulting policy. We implement multiple ways of combining GFNs and $Q$s, referring to the general idea as QGFN: taking a greedy with probability $p$ ($p$-greedy QGFN), restricting the agent to the top $1 - p\%$ of actions ($p$-quantile QGFN), and restricting the agent to actions whose estimated value is at least a fraction $p$ of the best possible value ($p$-of-max QGFN).

We chose to show several variants of QGFN in this paper, because they all rely on the same principle, learning $Q$, but have different properties, which lead to better or worse behavior in different tasks.

For example, pruning too aggressively on a task with a small number of actions is harmful. We also hope that by showing such a diversity of combinations of $P_F$ and $Q$, we encourage future work that combines GFNs and RL methods in novel and creative ways.

We also analyzed why our method works. We showed that the learned action-value $Q$ is predictive and helpful in avoiding actions that have high probability under $P_F$ but lower expected reward. Even when $Q$ predictions are not accurate, e.g. because we sample from a different policy than the one which $Q$ models, they provide a helpful lower bound that facilitates controllable greediness.

Our analysis suggests that at training time, QGFN works because it helps the agent to "waste" less time and capacity modeling low-reward objects, and that conversely the policy family that QGFN learns is able to sample more distinct high-reward objects given the same budget. In this sense, QGFN benefits from the advantages of both the GFlowNet and "Hard"-RL frameworks.

**What didn't work**    The initial stages of this project were quite different. Instead of combining RL and GFNs into one sampling policy, we instead trained two agents, a GFN and a DQN. Since both are off-policy methods we were hoping that sharing "greedy" DQN data with a GFN would be fine and make GFN better on high-reward trajectories. This was not the case, instead, the DQN agent simply slowed down the whole method–despite trying a wide variety of tricks, see §C.

**Limitations**    Because we train two models, our method requires more memory and FLOPs, and consequently takes more time to train compared to TB (as shown in Table 3). QGFN is also sensitive to how well $Q$ is learned, and as we've shown $n$-step returns are crucial for our method to work. In addition, although the problems we tackle are non-trivial, we do not explore the parameter and compute scaling behaviors of the benchmarked methods.

**Future work**    We highlight two straightforward avenues of future work. First, there probably exist more interesting combinations of $Q$ and $P_F$ (and perhaps $F$), with different properties and benefits. Second, it may be interesting to further leverage the idea of pruning the action space based on $Q$, forming the basis for some sort of constrained combinatorial optimization. By using $Q$ to predict some expected property or constraint, rather than reward, we could prune some of the action space to avoid violating constraints, or to keep some other properties below some threshold (e.g. synthesizability or toxicity in molecules).

Finally, we hope that this work helps highlight the differences between RL and GFlowNet, while adding to the literature showing that these approaches complement each other well. It is likely that we are only scratching the surface of what is possible in combining these two frameworks.

# Acknowledgements

The bulk of this research was done at Valence Labs as part of an internship, using computational resources there. This research was also enabled in part by computational resources provided by Calcul Québec, Compute Canada and Mila. Academic authors are funded by their respective academic institution, Fonds Recherche Quebec through the FACSAcquity grant, the National Research Council of Canada and the DeepMind Fellowships Scholarship.

The authors are grateful to Yoshua Bengio, Moksh Jain, Minsu Kim, and the Valence Labs team for their feedback, discussions, and help with baselines.

# Author Contributions

The majority of the experimental work, code, plotting, and scientific contributions were by EL, with support from EB. SL helped run some experiments, baselines and plots. The project was supervised by EB, and DP and LP provided additional scientific guidance. Most of the paper was written by EB. DP, LP, and EL contributed to editing the paper.

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

# A   Analysing $p$-greedy

Consider the bandit setting where trajectories are 1 step and just consist in choosing a terminal state. Let $p_G(s) = R(s)/Z$. Let $0 < p < 1$, then with $\mu(s'|s) = (1-p)P_F(s'|s) + p\mathbb{I}[s' = \arg\max Q(s,s')]$, assuming there is only a single argmax $s^*$, then $p_\mu(s) = (1-p)R(s)/Z + p\mathbb{I}[s = \arg\max R(s)]$. This means that for every non-argmax state, $p_\mu(s) = (1-p)p_G(s) < p_G(s)$. We get that $\mathbb{E}_\mu[R] > \mathbb{E}_G[R]$:

$$\mathbb{E}_\mu[R] - \mathbb{E}_G[R] = \sum_s p_\mu(s)R(s) - \sum_s p_G(s)R(s) \tag{6}$$

$$= (p + (1-p)R(s^*)/Z - R(s^*)/Z)R(s^*) + \sum_{s \neq s^*}(1-p)R(s)^2/Z - R(s)^2/Z \tag{7}$$

$$= pR(s^*) - pR(s^*)^2/Z + \sum_{s \neq s^*}(-p)R(s)^2/Z \tag{8}$$

$$= p/Z\left(R(s^*)Z - R(s^*)^2 - \sum_{s \neq s^*}R(s)^2\right), \quad Z = \sum_s R(s) \tag{9}$$

$$= p/Z\left(R(s^*)[\sum_s R(s)] - R(s^*)^2 - \sum_{s \neq s^*}R(s)^2\right) \tag{10}$$

$$= p/Z\left(R(s^*)[\sum_s R(s)] - \sum_s R(s)^2\right) \tag{11}$$

$$= p/Z\left(\sum_s R(s^*)R(s) - R(s)^2\right) \tag{12}$$

$$\tag{13}$$

since $R(s^*) > R(s)$ and both are positive then $R(s^*)R(s) > R(s)^2$ thus the last sum is positive. All other terms are positive, therefore $\mathbb{E}_\mu[R] - \mathbb{E}_G[R] > 0$.

In the more general case, we are not aware of a satisfying closed form, but consider the following exercise.

Let $m(s,s') = \mathbb{I}[s' = \arg\max Q(s,s')]$. Let $F'$ be the "QGFN flow" which we'll decompose as $F' = F_G + F_Q$ where we think of $F_G$ and $F_Q$ as the GFN and Q-greedy contributions to the flows. Then:

$$F'(s) = \sum_{z \in \text{Par}(s)} F'(z)((1-p)P_F(s|z) + pm(z,s)) \tag{14}$$

$$= \sum_z F'(z)(1-p)P_F(s|z) + \sum_z F'(z)pm(z,s) \tag{15}$$

$$= \sum_z F_G(z)(1-p)P_F(s|z) + F_Q(z)(1-p)P_F(s|z) + \sum_z F'(z)pm(z,s) \tag{16}$$

$$= (1-p)F_G(s) + \sum_z F_Q(z)\mu(z|s) + F_G(z)pm(z,s) \tag{17}$$

Recall that $p(s) \propto F(s)$. Intuitively then, the probability of being in a state is reduced by a factor $(1-p)$, but possibly increased by this extra flow that has two origins. First, flow $F_Q$ carried over by $\mu$, and second, new flow being "stolen" from $F_G$ from parents when $m(z,s) = 1$, i.e. when $s$ is the argmax child.

This suggests that flow (probability mass) in a $p$-greedy QGFN is smoothly redirected towards states with locally highest reward from ancestors of such states. Conversely, states which have

many ancestors for which they are not the highest rewarding descendant will have their probability diminished.

# B  Additional experiments and analyses

In this section, we provide additional experiments to support our main findings. We explore the use of $Q$ functions from different behavior policies, assess various QGFN inference variants, examine QGFN variants trained with alternative objectives, and investigate the effects of weight sharing in QGFN models.

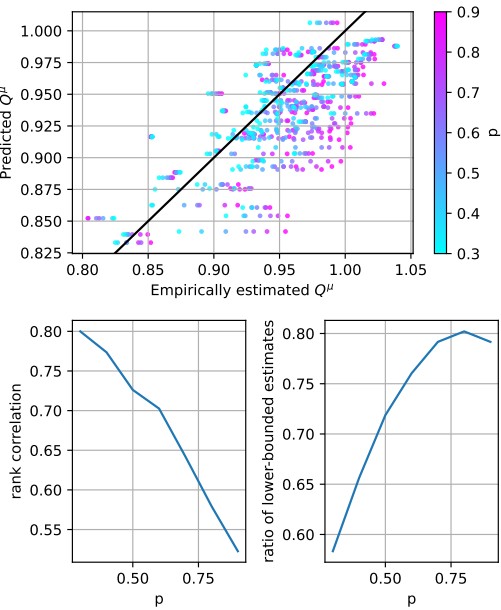

Figure 10: What happens to $Q(s, a)$ when changing $p$? We show here that while the rank correlation between $Q$ and the empirically estimated expected reward $\hat{Q}^{\mu_p}$ goes down when changing $p$, $Q$ remains a useful estimate in that it mostly lower bounds $\hat{Q}^{\mu}$. This means that, at worst, pruning based on the "wrong" $Q$ & $p$ combination drops some high-reward objects, but does not introduce more lower-reward objects.

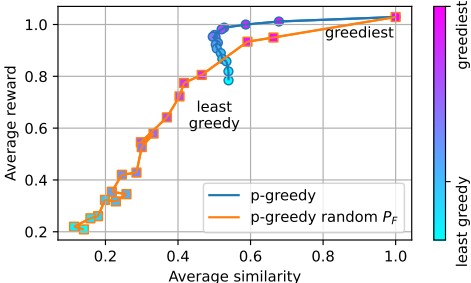

Figure 11: Comparison of trained $P_F$ vs. untrained $P_F$ with a trained $Q$ during inference on fragment-based molecule task

Table 2: Fragment-based molecule task: reward and diversity of independently trained baseline models using a trained $Q$. The $p$ values for $p$-greedy, $p$-of-max, and $p$-quantile QGFN are set at 0.4, 0.9858, and 0.93, respectively.

| VARIANT | TB | | SUBTB | |
|---|---|---|---|---|
| | REWARD | DIVERSITY | REWARD | DIVERSITY |
| BASELINE | 0.780±0.003 | **0.545±0.002** | 0.716±0.006 | **0.513±0.003** |
| P-GREEDY | 0.936±0.009 | 0.589±0.018 | 0.921±0.008 | 0.589±0.019 |
| P-OF-MAX | **0.953±0.004** | 0.545±0.026 | **0.939±0.003** | 0.536±0.023 |
| P-QUANTILE | 0.935±0.007 | 0.545±0.010 | 0.911±0.009 | 0.526±0.006 |

## B.1 Using $Q$ from different behavior policies

Another approach we explored involves using a trained $Q$ function during inference that was trained on an entirely different behavior policy. Similarly, we apply the QGFN algorithm at each state of sampling trajectories during inference, but with a key difference: it is directly applied to a baseline model that has been trained independently. This approach aims to examine if a previously trained $Q$, when used in a different training setup but the same task, can guide independently trained models that may not perform as well during training but, with this assistance, can achieve significantly better results at inference. For instance, as shown in Figure 2, the samples generated by SubTB average around 0.7 rewards throughout training. However, using the trained $Q$ as a greediness signal during inference allows us to discover samples with significantly higher rewards. Table 2 details the effects of applying $Q$ during inference on independently trained baseline models for 1000 samples post-training of 5 seeds.

## B.2 Trying other QGFN variants at inference

Since the only cost to trying to different "inference" variants of QGFN is to code them, we do so out of curiosity. We show the reward/diversity trade-off curves of these variants in Fig. 12, and include $p$-of-max as a baseline variant. As in Fig. 9 we take 512 samples for each point in the curves (except for MCTS which is more expensive). We try the following:

- $p$-thresh, mask all actions where $Q(s, a) < p$;
- soft-Q, not really QGFN, but as a baseline simply taking softmax$(Q/T)$ for some temperature $T$, which is varied as the greediness parameter;
- soft-Q [0.5], as above but mixed with $P_F$ with a factor $p = 0.5$ (i.e. $p$-greedy, but instead of being greedy, use the soft-$Q$ policy);
- GFN-then-Q, for the first $Np$ steps, sample from $P_F$, then sample greedily (where $N$ is the maximum trajectory length);
- MCTS, a Monte Carlo Tree Search where $P_F$ is used as the expansion prior and $\max_a Q(s, a)$ as the value of a state. Since this is a different sampling method, we run MCTS for a comparable amount of time to other variants, getting about 350 samples, and report the average reward and diversity.

We note that these are all tried using parameters from a pretrained $p$-greedy QGFN model. It may be possible for these variants to be much better at inference if the $Q$ used corresponded to the sampling policy.

## B.3 QGFN variants with different objective

To demonstrate the robustness of QGFN variants, we explore QGFN with different learning objectives such as SubTB in addition to the TB objective used throughout our experiments. We use the same hyperparameters, except the $p$ values (p-greedy 0.4, p-of-max 0.7, p-quantile 0.7), listed in Table 4 and run the experiments on fragment-based molecule generation. The results are shown in Figure 13 and Figure 14.

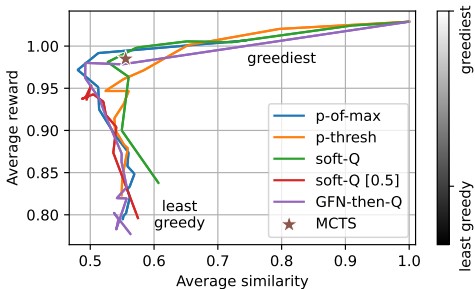

Figure 12: Trying other possible QGFN variants.

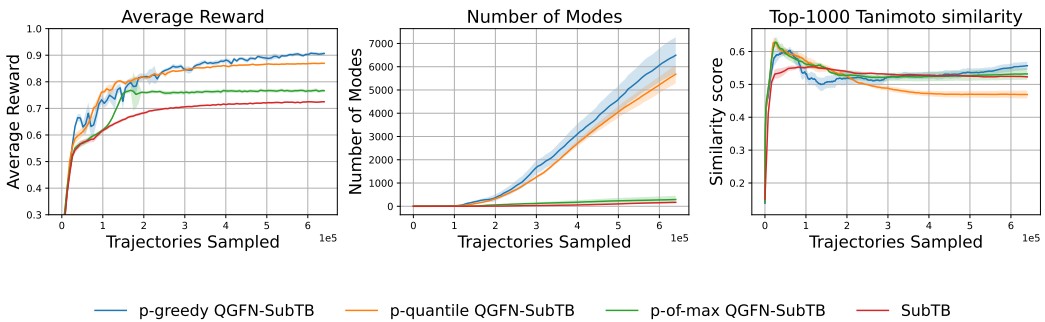

Figure 13: QGFN variants on learning objectives SubTB on Fragment-based molecule task

## B.4 Exploring weight sharing in QGFN

We explore the impact of weight sharing between $P_F$ and $Q$, as they learn from the same environments and training samples. This sharing learning approach could improve in efficiency and performance. Fig. 15 shows the impact of weight sharing on $p$-greedy QGFN, specifically focusing on sharing parameters across various layers of the graph attention transformer in the fragment-based task.

Unfortunately, naively summing the GFlowNet loss and the $Q$-Learning loss does not yield any improvements, and instead slows down learning. This may be due to several factors; most likely, interference between $Q$ and $P_F$, and a very different scales of the loss function may induce undesirable during training. A natural next stop would be to adjust the relative magnitude of the gradients coming from each loss, and to consider different hyperparameters (perhaps a higher model capacity is necessary), but we leave this to future work. Further exploration in this area could provide additional insights and potentially reduce training complexity.

## B.5 Training Time and Inference Time Comparison

In addition to performance, we investigate the training time and inference time for TB and $p$-greedy QGFN. All experiments for this comparison were conducted on NVIDIA V100 GPUs. The results are reported in Table 3.

Table 3: Training and inference time comparison between TB and $p$-greedy QGFN.

|  | TB | QGFN ($p$-greedy) |
| --- | --- | --- |
| Training time (10,000 training iterations) | 3 hours, 25 minutes | 6 hours, 20 minutes |
| Inference time (1,000 samples) | 2 min, 44 sec | 2 min, 21 sec |

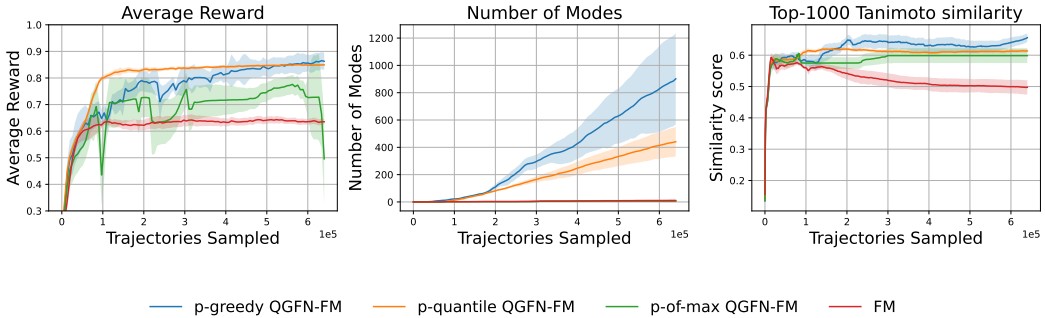

Figure 14: QGFN variants on learning objectives FM (Flow Matching) on Fragment-based molecule task

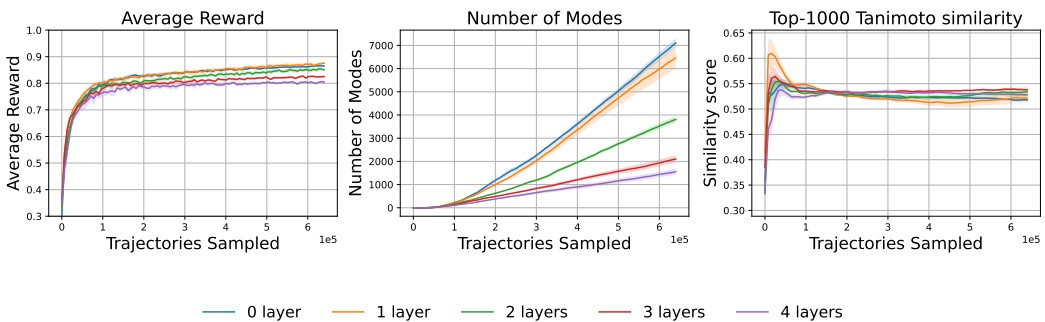

Figure 15: Weight sharing in $p$-greedy QGFN (p=0.4) with different layers

## C    Experiments that did not work

Several approaches were attempted prior to developing QGFN. These approaches involved sampling from both GFN and DQN independently and learning from the shared data. The key strategies explored include:

- **Diverse-Based Replay Buffer**: This method stores batches of trajectories in the replay buffer and samples them based on a pairwise Tanimoto similarity threshold. It aims to diversify the experience replay during training.
- **Adaptive Reward Prioritized Replay Buffer**: This strategy stores batches of trajectories in the replay buffer based on the rewards of the samples. In addition, we dynamically adjusts the sampling proportion between GFN and DQN based on the reward performance of the trajectories.
- **Weight Sharing**: This involves sharing weights between GFN and DQN to potentially enhance the learning and convergence of the models.
- **Pretrained-Q for Greedier Actions**: This method uses a pretrained DQN model for sampling trajectories, helping the GFN to be biased towards greedier actions in the early learning stages.
- $n$**-step returns**: As per QGFN, using more than 1-step temporal differences can accelerate temporal credit assignment. This on its own is not enough to solve the tasks used in this work.

Fig 16 shows the performance of these approaches evaluated on fragment based molecule generation task.

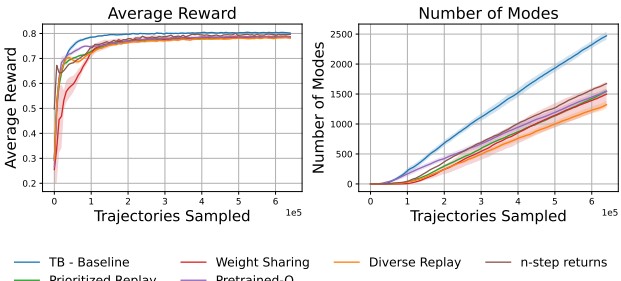

Figure 16: Fragment-based molecule generation task; we showcase the performance of QGFN's predecessor, which failed to beat baselines regardless of our attempts to improve it.

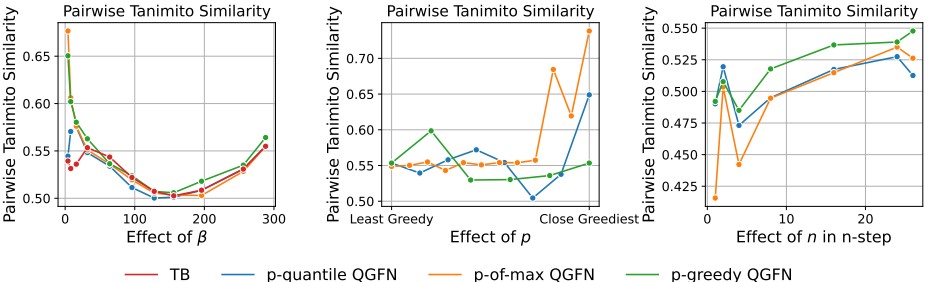

Figure 17: Pairwise Tanimoto Similarity scores assessing the impacts of $\beta$, training parameters $p$ and $n$ in the fragment task. *Left:* An increase in $\beta$ initially decreases sample similarities, followed by a gradual increase in similarity as the models get greedier through $\beta$. *Center:* Increase greediness does not always correlate with sample similarity trade-offs with QGFN, but at peak greediness, similarity scores rebound. *Right:* Increasing $n$ increases similarity among models.

# D   Full Algorithm

In this section, we show the detailed implementation of the QGFN algorithm with different variants in Algorithm 1. For inference, the trained models $P_F$ and $Q$ can be loaded to sample trajectories.

**Algorithm 1** QGFN: Full training algorithm details

**Require:** Reward function $R : \mathcal{X} \to \mathbb{R}_{>0}$, batch size $M$, Initialize models $P_F$ with parameters $\theta$, $Q(s, a)$ with parameters $\theta'$, greediness parameter $p \in [0, 1]$, training iterations $I$

1: **For $p$-greedy QGFN:**

$$\mu(s'|s) = (1 - p)P_F(s'|s) + p\mathbb{I}[s' = \arg\max_i Q(s, i)] \tag{18}$$

2: **For $p$-quantile QGFN:**

$$\mu(s'|s) \propto P_F(s'|s)\mathbb{I}[Q(s, s') \geq q_p(Q, s)] \tag{19}$$

where $q_p(Q, s)$ is the $p$-quantile of $Q$ over actions at state $s$.

3: **For $p$-of-max QGFN:**

$$\mu(s'|s) \propto P_F(s'|s)\mathbb{I}[Q(s, s') \geq p\max_i Q(s, i)] \tag{20}$$

4: **for** for training iteration $i$ in $I$ **do**
5:     **for** each new trajectory $\tau_j$ from $\tau_1$ to $\tau_M$ **do**
6:         Start $\tau_j$ at $s_0$
7:         **while** $s_t$ is not terminal **do**
8:             Sample $s_{t+1}$ from $\mu(s_{t+1}|s_t)$ based on current policy
9:             Update $t \leftarrow t + 1$
10:     Compute trajectory balance loss for $P_F$: $\sum_j \mathcal{L}_{\text{TB}}(\tau_j)$
11:     Compute MSE n-step loss for Q-network:

$$\mathcal{L}_{\text{Q}} = \mathbb{E}_{(s_t, a_t)}\left[\left(Q(s_t, a_t) - G_t^{(n)}\right)^2\right]$$

where the n-step return $G_t^{(n)}$ is defined as:

$$G_t^{(n)} = \sum_{k=0}^{n-1} \gamma^k r_{t+k} + \gamma^n \max_{a'} Q(s_{t+n}, a')$$

12:     Update $\theta$ using $\nabla_\theta \mathcal{L}_{\text{TB}}$;
13:     Update $\theta'$ using $\nabla_{\theta'} \mathcal{L}_{\text{Q}}$;

## E   Experiment details: Fragment-based molecule generation

In this section, we give the hyperparameters used for each of our experiments in Tables 4, and Table 5.

| Parameter | Value |
|---|---|
| Batch size | 64 |
| Number of steps | 10,000 |
| Optimizer | Adam |
| Number of Layers | 4 |
| Hidden Dim. Size | 128 |
| Number of Heads | 2 |
| Positional Embeddings | Rotary |
| Reward scaling $\beta$ in $R^\beta$ | 32 |
| Learning rate | $1 \times 10^{-4}$ |
| $Z$ Learning rate | $1 \times 10^{-3}$ |

Table 4: Hyperparameters and specifications of the Graph Attention Transformer used across all models in Fragment-based molecule generation.

| Parameter | Value |
|---|---|
| Objective function | TB |
| $p$-greedy | 0.4 |
| $p$-quantile | 0.8 |
| $p$-of-max | 0.91 |
| cosine scheduler for $p$ | 1500 steps |
| Model architecture | Graph Attention Transformer |
| $n$-step | 25 |
| dqn $\tau$ | 0.95 |

Table 5: Model-specific parameters for QGFN in Fragment-based molecule generation.

In Table 1, the $p$ values for $p$-greedy, $p$-of-max, and $p$-quantile QGFN are set to 0.4, 0.9858, and 0.93, respectively. These values are selected based on Figure 9. The $p$ for $p$-of-max is chosen from `np.linspace(0.9, 0.999, 16)`, with 0.9858 being one of these values. Similarly, for $p$-quantile, 0.93 corresponds to the second-to-last value from `np.linspace(0, 1, 16)`. Meanwhile, the 0.4 for $p$-greedy is selected from `np.linspace(0, 1, 11)`. For LSL-GFN, the chosen $\beta$ is 78, selected from `np.linspace(64, 128, 65)`.

In our experimental setup, we follow the exact environment specifications and implementations detailed in Malkin et al. [2022a] with the proxy model, used for evaluating molecules, provided by Bengio et al. [2021b]. The architecture of the GFlowNet models is based on a graph attention transformer [Veličković et al., 2017]. We set a reward threshold of 0.97 to define modes, with a pairwise Tanimoto similarity criterion of less than 0.65. RDKit [Landrum, 2013] is used to compare pairwise Tanimoto similarity.

To follow closely the original implementation of LSL-GFN described in Kim et al. [2023a], we use $\beta \sim U^{[1,64]}$, where $U$ denotes a uniform distribution. Additionally, we define a simple Multi-layer Perceptron with a hidden size of 256 as the learnable scalar-to-scalar function for the LSL-GFN. For A2C, we use a learning rate of $1 \times 10^{-4}$, a training epsilon of $1 \times 10^{-2}$, and an entropy regularization coefficient of $1 \times 10^{-3}$. For our SubTB baseline we use SubTB(1), i.e. all trajectories are averaged with equal weight.

To maintain consistency, the graph attention transformer was used as the model for MunDQN. We sampled 64 trajectories and stored them as transitions in a prioritized replay buffer of size 1,000,000. We then sampled 4096 transitions from the replay buffer to calculate the loss. The Munchausen parameters of 0.10 is selected from $\{0.10, 0.15\}$, an $l_0$ of -2500 and a soft update of $\tau = 0.1$ is used in our experiments. All other parameters are same as the original MunDQN paper in Tiapkin et al. [2023].

We also ran an SAC [Haarnoja et al., 2018] baseline with different $\alpha$ values of 0.5, 0.7, 0.2, along with autotuning, and a $\gamma$ value of 0.99, but we were unable to get it to discover more than 50 modes for the same amount of training iterations and mini-batch sizes.

### E.1 QGFN hyperparameters:

For all variants of QGFN, we employed a grid-search approach for hyperparameter tuning, with a focus on the parameters $p$ and $n$. Similarly, graph attention transformer is initialized as the $Q$. In the $p$-greedy QGFN variant, we selected a value of $0.4$ for $p$ from the set $\{0.2, 0.4, 0.6, 0.8\}$, and chose an $n$ of 25 from the set $\{1, 2, 4, 8, 16, 24, 25, 26\}$. For the $p$-of-max QGFN variant, a $p$ of 0.91 was chosen from $\{0.2, 0.4, 0.6, 0.8, 0.9, 0.91, 0.93, 0.95\}$, with $n$ again set at 25. To ensure stability throughout the training process, we applied cosine annealing with a single-period cosine schedule over 1500 steps. An additional threshold parameter of $1 \times 10^{-5}$ was applied to the condition $p \max_i Q(s, i) >$ threshold, to prevent the initial training phase from masking actions with very small values. We also introduced a clipping of the Q-value to a minimum of 0 to prevent instability during initial training. For the $p$-quantile QGFN, a $p$ of 0.8 was selected from $\{0.6, 0.7, 0.8, 0.9, 0.95\}$, with $n = 25$. For all variants of QGFN, the DQN employed had a $\tau$ of 0.95, and random action probability of $\epsilon$ was set to 0.1.

# F    Experiment details: QM9

| Parameter | Value |
|---|---|
| Batch size | 64 |
| Number of steps | 5,000 |
| Optimizer | Adam |
| Number of Layers | 4 |
| Hidden Dim. Size | 128 |
| Number of Heads | 2 |
| Positional Embeddings | Rotary |
| Reward scaling $\beta$ in $R^\beta$ | 32 |
| Learning rate | $1 \times 10^{-4}$ |
| $Z$ Learning rate | $1 \times 10^{-3}$ |

Table 6: Hyperparameters and specifications of the Graph Attention Transformer used across all models in QM9.

| Parameter | Value |
|---|---|
| Objective function | TB |
| $p$-greedy | 0.4 |
| $p$-quantile | 0.6 |
| $p$-of-max | 0.6 |
| cosine scheduler for $p$ | 1500 steps |
| Model architecture | Graph Attention Transformer |
| $n$-step | 29 |
| dqn $\tau$ | 0.95 |

Table 7: Model-specific parameters for QGFN in QM9.

In this experiment, we follow the setup described by Jain et al. [2023], but only use the HOMO-LUMO gap as a reward signal. The rewards are normalized to fall between [0, 1], although the gap proxy may range from [1,2]. As mentioned in Section 5, the modes are computed with a reward threshold of 1.10 and a pairwise Tanimoto similarity threshold of 0.70. We employ the same architecture for all models as used in the fragment-based experiments. The training models are 5,000 iterations with a mini-batch size of 64, and $\beta$ is set to 32. RDKit [Landrum, 2013] is used to compare pairwise Tanimoto similarity. We train A2C with random action probability 0.01 chosen from $\{0.1, 0.01, 0.001\}$ and entropy regularization coefficient 0.001 chosen from $\{0, 0.1, 0.01, 0.001\}$.Similar to the fragment-based molecule task, we initialized the graph attention transformer with the Munchausen parameter $\alpha$ set to 0.15, a prioritized replay buffer size of 1,000,000, and a soft update coefficient $\tau = 0.1$. We sample 64 trajectories and store them in the replay buffer, subsequently sampling 4096 transitions from this buffer. We used the other hyperparameters mentioned in the original MunDQN paper [Tiapkin et al., 2023].

## F.1    QGFN hyperparameters:

For all variants of QGFN, we employed an exhaustive grid-search approach for hyperparameter tuning, focusing on parameters $p$ and $n$. For $p$-greedy QGFN, we selected 0.4 for $p$ from $\{0.2, 0.4, 0.6\}$ and 29 for $n$ from $\{11, 29, 30\}$. For $p$-of-max QGFN, we chose 0.6 for $p$ from $\{0.3, 0.4, 0.5, 0.6, 0.7, 0.8, 0.9\}$ and 29 for $n$ from $\{28, 29, 30\}$. For $p$-qunatile QGFN, we selected 0.6 for $p$ from $\{0.5, 0.6, 0.7, 0.8\}$ and 29 for $n$ from $\{11, 27, 28, 29\}$. Similarly to the fragment task, we implemented an additional threshold of $1 \times 10^{-3}$ to $p \max_i Q(s, i) >$ threshold and clipped Q-values to a minimum of 0 for stability during initial training. Cosine annealing over 1500 steps is used for all variants.

# G    Experiment details: RNA-binding task

| Parameter | Value |
|---|---|
| Batch size | 64 |
| Number of steps | 5,000 |
| Optimizer | Adam |
| Number of Layers | 4 |
| Hidden Dim. Size | 64 |
| Number of Heads | 2 |
| Positional Embeddings | Rotary |
| Reward scaling $\beta$ in $R^\beta$ | 8 |
| Learning rate | $1 \times 10^{-4}$ |
| $Z$ Learning rate | $1 \times 10^{-2}$ |

Table 8: Hyperparameters and specifications of the Sequence Transformer used across all models in RNA-binding task.

| Parameter | Value |
|---|---|
| Objective function | TB |
| $p$-greedy | 0.4 |
| $p$-quantile | 0.25 |
| $p$-of-max | 0.9 |
| stepwise scheduler for $p$ | 500 steps |
| Model architecture | Sequence Transformer |
| $n$-step | 13 |
| dqn $\tau$ | 0.95 |

Table 9: Model-specific parameters for QGFN in RNA-binding task.

We follow the setup of Jain et al. [2022] but using the task introduced in Sinai et al. [2020]. We use a sequence transformer [Vaswani et al., 2017] architecture with 4 layers, 64-dimensional embeddings, and 2 attention heads. The training for this task is 5000 iterations over mini-batch sizes of 64. The reward scaling parameter $\beta$ is set to 8 and a learning rate of $1 \times 10^{-4}$ and $1 \times 10^{-2}$ for $\log Z$. In this task, $\beta \sim U^{[1,16]}$ is used for LSL-GFN. Following the approach described by Sinai et al. [2020], the modes are predefined from enumerating the entire RNA landscape for L14RNA1 and L14RNA1+2 to identify local optimal through exhaustive search. ViennaRNA [Lorenz et al., 2011] is used to provide the RNA binding landscape. For MunDQN, Munchausen parameter $\alpha$ set to 0.15, a prioritized replay buffer size of 800,000, and a soft update coefficient $\tau = 0.1$. We sample 16 trajectories and store them in the replay buffer, subsequently sampling 1024 transitions from this buffer. We used the other hyperparameters mentioned in the original MunDQN paper [Tiapkin et al., 2023].

## G.1    QGFN hyperparameters:

For all QGFN variants, we used grid-search for tuning hyperparameters $p$ and $n$. We used $n = 13$ from the set $12, 13, 14$. For $p$-greedy QGFN, $p = 0.4$ was chosen from $\{0.2, 0.4, 0.6, 0.8\}$. For $p$-of-max QGFN, $p = 0.9$ was selected from $\{0.6, 0.7, 0.8, 0.9\}$. In $p$-quantile QGFN, we tried $p = 0.25$ and $0.50$, but neither performed well due to small action spaces. A stepwise scheduler set at 500 steps is applied to $p$-of-max QGFN.

## H    Experiment details: Bit sequence generation

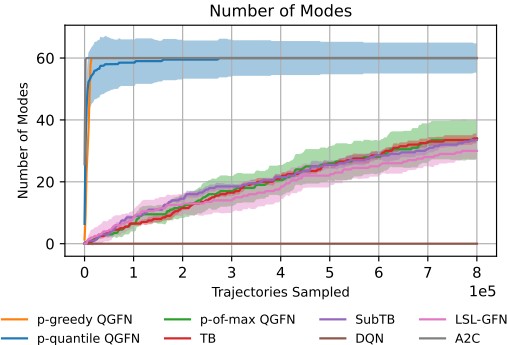

Figure 18: Bit sequence generation, $k = 4$.

| Parameter | Value |
|---|---|
| Batch size | 16 |
| Number of steps | 50,000 |
| Optimizer | Adam |
| Number of Layers | 3 |
| Hidden Dim. Size | 64 |
| Number of Heads | 2 |
| Positional Embeddings | Rotary |
| Reward scaling $\beta$ in $R^\beta$ | 3 |
| Learning rate | $1 \times 10^{-4}$ |
| $Z$ Learning rate | $1 \times 10^{-2}$ |

Table 10: Hyperparameters and specifications of the Sequence Transformer used across all models in bit sequence generation.

| Parameter | Value |
|---|---|
| Objective function | TB |
| $p$-greedy | 0.4 |
| $p$-quantile | 0.25 |
| $p$-of-max | 0.3 |
| stepwise scheduler for $p$ | 500 steps |
| Model architecture | Sequence Transformer |
| $n$-step | 120 |
| dqn $\tau$ | 0.95 |

Table 11: Model-specific parameters for QGFN in bit sequence generation.

The bit sequence generation task follows the same environmental setup as Malkin et al. [2022a] with $\beta$ value as 3. We generate $|M| = 60$ reference sequences by randomly combining $m = 15$ symbols from an initial vocabulary $H = \{00000000, 11111111, 11110000, 00001111, 00111100\}$.

Beyond the original auto-regressive generative framework in Malkin et al. [2022a], we generate sequences in a prepend-apppend fashion motivated by Shen et al. [2023]. We use a sequence transformer [Vaswani et al., 2017] architecture for all experiments with rotary position embeddings, 3 hidden layers with hidden dimension 64 across 8 attention heads. All methods were trained for 50,000 iterations with a minibatch size of 16. For trajectory balance, we use a learning rate of of $1 \times 10^{-4}$

for the policy parameters and $1 \times 10^{-3}$ for $\log Z$. For SubTB, we use the same hyperparameters as in Madan et al. [2023]. For LSL-GFN, we use $\beta \sim U^{[1,6]}$, where $U$ denotes a uniform distribution.

### H.1 QGFN hyperparameters:

For all variants of QGFN, we did a grid-search approach for hyperparameter tuning for $p$ and $n$. For k=1 where actions are limited to $\{0,1\}$, we set $n$ at 120, selected from $\{30, 60, 90, 120\}$, across alla QGFN variants. For the p-greedy QGFN, we chose $p = 0.4$; for the $p$-of-max QGFN, $p$ was set to 0.3, with cosine annealing applied at 500 steps. In the $p$-qunatile QGFN, we tested $p$ values of 0.25 and 0.50, but neither achieved good performance due to the binary nature of the action space.

## I Experiment details: Graph combinatorial optimization problems - maximum independent set (MIS)

As an additional task, we explore graph combinatorial optimization, specifically the maximum independent set (MIS) problem mentioned in Zhang et al. [2023]. We directly used the codebase shared by Zhang et al. [2023] and report the performance at test time in Table 12.

| Method | Small - Metric Size | Small - Top 20 |
| --- | --- | --- |
| FL | 18.20 | 18.72 |
| FL - QGFN (p-greedy) | 18.21 | **19.06** |
| FL - QGFN (p-quantile) | 18.20 | 18.75 |
| FL - QGFN (p-of-max) | **18.26** | 18.74 |

Table 12: Comparison of different methods on small graphs: metric size and top 20 metrics.

We used the same parameters as in the fragment-based molecule generation experiments. At test time, we applied the p-greedy, p-quantile, and p-of-max sampling strategies respectively for different methods. Note that the reported performance might not reflect the best achievable results on this task, as we did not explore different hyperparameter settings.

## J Compute Resources

All of our experiments were conducted using A100 and V100 GPUs. For the fragment-based task, we used 8 workers on A100 GPUs, and it ran in less than 4 hours. For RNA, we used 4 workers, and it completed in less than 4 hours. For QM9, we used 0 workers, and it finished in less than 9 hours. For Bit sequence, we used 8 workers, and it ran in less than 24 hours.

## K Broader Impacts

The research conducted in this work is not far removed from practical applications of generative models. As such, we acknowledge the importance of considering safety and alignment in applications closely related to this work such as drug discovery, material design, and industrial optimization. We believe that research on GFlowNets may lead to models that better generalize, and in the long run may be easier to align. Another important application of GFlowNets is in the causal and reasoning domains; we believe that improving in those fields may lead to easier to understand and safer models.

