# OpenReview forum: "QGFN: Controllable Greediness with Action Values"
_NeurIPS.cc/2024/Conference — NeurIPS 2024 poster_

### Official Review · Reviewer_eVPz · 2024-06-17

**Soundness:** 3
**Presentation:** 3
**Contribution:** 3
**Rating:** 5
**Confidence:** 3

**Summary:**

The paper "QGFN: Controllable Greediness with Action Values" introduces a novel approach to enhance Generative Flow Networks (GFNs) by incorporating action-value estimates (Q-values) to control the greediness of sampling policies. This method, called QGFN, includes three variants—p-greedy, p-quantile, and p-of-max—each designed to balance the generation of high-reward samples with the maintenance of diversity. Through comprehensive experiments on tasks like molecule generation and RNA design, the authors demonstrate that QGFN significantly improves the generation of high-utility samples while preserving diversity, providing a practical and effective solution for deploying safe and helpful LLMs.

**Strengths:**

- **Innovative Approach**: The introduction of Q-values to control the greediness of sampling policies in Generative Flow Networks (GFNs) is a novel and creative solution to the challenge of balancing high-reward sample generation with diversity.
- **Comprehensive Evaluation**: The paper includes thorough experiments across various tasks, such as molecule generation and RNA design, providing strong empirical evidence for the effectiveness of QGFN.
- **Ablation Study**: The paper provides in-depth ablation studies on the hyperparameters of GFNs.

**Weaknesses:**

- **QGFN variants matter**. This paper does not provide a method to select different QGFN variants. Different variants of QGFN have very different performances in different environments. In Section 5, the authors claim that p-of-max is suitable for small action spaces, and p-quantile is suitable for large action spaces. What if different states have different action spaces? For example, in the graph combinatorial optimization problems [1], earlier states have a much larger action space than later states, and the action space size will change (decrease) during the sampling. I think compared with a fixed action space, this setting probably requires the use of different QGFN variants on different states during the sampling.
- **Lack of more complex environment**. Therefore, I wonder about the QGFN's performance on graph combinatorial optimization problems, such as MIA. [1]
- There are some missing recent works that deal with sampling high-reward candidates with diversity, such as [2, 3]
- I am happy to raise my scores if my concerns are resolved.


[1] Zhang, D., Dai, H., Malkin, N., Courville, A., Bengio, Y., & Pan, L. (2023). Let the flows tell: Solving graph combinatorial optimization problems with gflownets. arXiv preprint arXiv:2305.17010.

[2] Chen, Y., & Mauch, L. (2023). Order-Preserving GFlowNets. arXiv preprint arXiv:2310.00386.

[3] Jang, H., Kim, M., & Ahn, S. (2023). Learning Energy Decompositions for Partial Inference of GFlowNets. arXiv preprint arXiv:2310.03301.

**Questions:**

See **Weaknesses**

**Limitations:**

See **Weaknesses**

---

> ### Author Rebuttal · Authors · 2024-08-07
>
> Thank you for your feedback. We appreciate the time and effort you put into reviewing our work. Below, we address your questions and comments in detail. We hope this clarifies our approach and findings.
>
> > QGFN variants matter. This paper does not provide a method to select different QGFN variants. Different variants of QGFN have very different performances in different environments. In Section 5, the authors claim that p-of-max is suitable for small action spaces, and p-quantile is suitable for large action spaces. What if different states have different action spaces? For example, in the graph combinatorial optimization problems [1], earlier states have a much larger action space than later states, and the action space size will change (decrease) during the sampling...
>
> This is a great question, and as the reviewer points out, we find that the optimal variant tends to be a function of the action space. Although not always optimal, the $p$-greedy variant consistently outperforms its GFN counterpart in finding modes and is a safe choice to default to that is less dependent on the action space.
>
> We also would like to highlight that a variant such as $p$-of-max QGFN will select a variable number of actions (any action whose $Q$ value is at least a fraction $p$ of the best action's $Q$ value), and might be a suitable choice for varying action sizes. It is also likely that more appropriate mixtures exist. Additionally, note that in both the fragment-based molecule generation task and atom-based molecule generation task (QM9), the size of the action space is highly variable as it depends on the number of generated fragments/atoms at each timestep.
>
> > Lack of more complex environment. Therefore, I wonder about the QGFN's performance on graph combinatorial optimization problems, such as MIA. [1]
>
> While we feel that the fragment-based molecule task is complex (Figure 2; $|\mathcal{X}|\approx 10^{100}$), we address the concern by training QGFN on the MIS codebase [1] with default hyperparameters. From superficial experimentation over the last few days, we show we can improve performance upon the baseline model.
>
> | METHOD               | Small - Metric Size | Small - Top 20 |
> |----------------------|---------------------|----------------|
> | FL                   | 18.20              | 18.72         |
> | FL - QGFN (p-greedy) | 18.21              | **19.06**         |
> | FL - QGFN (top-p)    | 18.20              | 18.75         |
> | FL - QGFN (high-p)   | **18.26**              | 18.74         |
>
> Table A: Max independent set experimental results on small dataset (200 to 300 vertices) using FL-GFlowNets (Pan et al., 2023). We sample 20 solutions for each graph configuration and get the average and top results.
>
> Due to the limited time for training during the rebuttal period, we anticipate that experimenting with different hyperparameters could yield even better performance. We will include these extended results in the revised paper for a more thorough evaluation.
>
> > There are some missing recent works that deal with sampling high-reward candidates with diversity, such as [2, 3]
>
> We appreciate that other GFN methods exist that increase the diversity-reward quality of sampling. However, our experimentation focuses on showcasing how QGFN can be easily integrated into the GFN framework across various tasks with improved performance. In other words, one _could_ train OP-QGFNs and LED-QGFNs. As shown in the above experimentation on MIS, we successfully used QGFN variants on FL-GFlowNets (Pan et al., 2023) and showed that they can adapt to varying action spaces dynamically and provide a boost in performance.

---

> > ### Comment · Reviewer_eVPz · 2024-08-09
> >
> > Thank you for the clarifications. However, I think the question, "How to choose the best QGFN variants given different environment" is still unsolved.  Given a state $s$ and the action space $A(s)$ at this state, there should be more detailed instruction on how to choose the QGFN variants and $p$. Regarding the best $p$, I have additional concerns below from the question from the first reviewer.
> >
> > From your rebuttal, the choice of $p$ in Table 1 is based on Figure 6. the chosen $p$ of p-greedy, p-of-max, and p-quantile QGFN are 0.4, 0.9858, and 0.93 respectively, which is very different from each other (and very strange numbers).  According to Figure 6, the p-of-max close to the "Greediest" ($p=1$) will have inferior performance than "Least Greedy" ($p=0$). Therefore, the choice of an appropriate $p$ is non-trivial. (Also, The color lines for the left two and right two subfigures seem to be inconsistent, and for the 3rd subfigures, no numbers on the x-axis. ) I think choosing training $p$ using Figure 6 is not appropriate.
> >
> > if you are given a trained $P_F$ and $Q$, I think you can enumerate different $p$s at inference to select the best $p$ to combine them together at inference. However, as I reread the paper, the different $p$s are chosen in training, I do not think it is acceptable to enumerate all different $p$ in training since it will be very computationally expensive to retrain the model for every $p$ and not fair to other methods.
> >
> > I temporarily decrease my score to 4. If the above concerns are resolved. I am willing to raise my scores again.

---

> > > ### Author Response · Authors · 2024-08-10
> > > **Response**
> > >
> > > Thank you for engaging with us. We apologize for a mistake in our rebuttal to bgQq, inference time $p$ values are chosen based on Figure 9, not Figure 6. Figure 6's left two figures use these inference-time $p$ values, while Figure 6's 3rd subplot show the effect of different (fixed) $p$ values used for training.
> > >
> > > We understand your perspective, but we believe it comes from us not properly communicating the facts rather than what appears to be (and would be very concerning!) _strange numbers_.
> > >
> > > Below are our detailed responses to your questions. We hope these address your concerns. We are happy to answer any further questions you may have.
> > >
> > > Let's start with the choice of $p$. We make two claims:
> > > - A, $p$ can be treated like a hyperparameter, set once during training,
> > > - B, $p$ (and $\mu$) can be changed after training to _even further_ improve the reward-diversity quality of samples.
> > >
> > > Figures 2,3,4,5, and 6 (3rd subplot) show A. Treat $p$ as a hyperparameter, employ standard hyperparameter search. Even in this setting, QGFN shows promise. We hope this clarifies our approach to the reviewer.
> > >
> > > In a desire to show how malleable our method is, we went further and claimed B. This is shown in Figures 6 (two left figures) and 9 and Table 1. Without any retraining, we can get even more performance out of the models by tweaking one single parameter, $p$.
> > >
> > > **What is this "Greediest" to "Least Greedy" spectrum?** We chose this as an axis rather than 0-1 because 0-1 makes little sense for p-of-max. Instead (and we wrote this in l290-292 but clearly need to make this more prominent in the text), we use 0.9-1 for p-of-max and 0-1 for p-quantile and p-greedy. Below 0.9, p-of-max in practice samples from $P_F$ so it would have been a waste of space to show the 0-0.9 values. To overlay all the results on a single plot, it made more sense to us to simply show the evolution of behavior as a function of "greediness" rather than put too many details and multiple overlapping axes. This in retrospect may have been a confusing choice for the reader, and we will change this to make it clearer.
> > >
> > > Next, while 0.9853 may seem unusual and arbitrary, it is computed during inference with $p$ chosen from `np.linspace(0.9, 0.999, 16)`. Similarly, 0.93 is the second to last value of `np.linspace(0, 1, 16)`.
> > >
> > > > I think choosing training $p$ using Figure 6 is not appropriate
> > >
> > > The 3rd subplot of Figure 6 does **not** show $p$ determined **after** training, it shows the result of a hyperparameter search for $p$. Why would it not be appropriate to chose a training hyperparameter from the result of this?
> > >
> > > > I do not think it is acceptable to enumerate all different $p$ in training
> > >
> > > We believe we perform standard hyperparameter search, $p$ is chosen beforehand before any training occurs. This means $p$ is treated like any other hyperparameter, like learning rate, or batch size. This is a common practice in method papers.
> > >
> > > We want to clarify that we are **not** adjusting $p$ during training based on inference performance. Claim B is entirely about what happens _after_ having trained models during inference. This distinction between Claims allows our approach to be computationally feasible and fair when compared to other methods.
> > >
> > > Note that for baselines, we _also_ do a hyperparameter search on greed-controlling parameters, notably $\beta$. Part of the results of this search are shown in Figure 6's left two subplots. This is fair standard practice.
> > >
> > > Finally on this point, we apologize for the color confusion of Figure 6; we will correct this in the revision.
> > >
> > > > there should be more detailed instruction on how to choose the QGFN variants and $p$
> > >
> > > We did not explicitly state this, as our focus is on showing the usefulness of the _existence_ of $Q$-$P_F$ mixtures, which we think is a valuable NeurIPS-level scientific contribution in itself, rather than advising on a specific approach.
> > >
> > > However, we understand that readers may be curious about which variant to use, and we reiterate what we write in our rebuttal, and will emphasize in our paper that starting with $p$-greedy (with $p=0.5$) is a reliable option.
> > >
> > > We speculate in our paper that variants have effects which partially depend on the action space because of the nature our tasks. These speculations are based on evidence, but instructions for the general case would require vastly more evidence, more compute, on many more types of action spaces than what we have. We hope to impress upon the reviewer that this would be worthy of an entirely different 9-page paper.
> > >
> > > We have introduced a novel method, shown that it works, and that it works because of the hypothesized mechanism, and went beyond that and showed that many variants of the idea are both sensible and capable of producing interesting results. We agree with the reviewer that this paper would be stronger if we had a general recipe, but providing such a comprehensive solution would be beyond the intended scope and contribution of this paper.

---

> > > > ### Comment · Reviewer_eVPz · 2024-08-13
> > > >
> > > > Thanks for your response. However, although $p$ can be seen as a hyperparameter in a standard hyperparameter search, it is still an additional hyperparameter compared with other common hyperparameters shared with all GFN methods, such as batch size, learning rate and $\beta$.  Also, for each iteration of search on $p$, one GFN network and one Q network needs to be trained, it is additional burden compared with GFN only. Comparing with modified GFN, such as local search GFN, LED-GFN, QGFN seems to be more computationally expensive by an additional Q, and the paper and the rebuttal do not provide a comparison in the experiments (although this paper cited local search GFN). In the rebuttal to the Reviewer bgQq, the authors said that Q function could help improve these modified GFN but no experiments except for FL-GFN, a relatively weaker baseline, either. Therefore, I maintain my current score.

---

> > > > > ### Author Response · Authors · 2024-08-13
> > > > > **Response**
> > > > >
> > > > > Thank you again for engaging with us. We'd like to reiterate that our research methodology aligns with standard practices in the ML field.
> > > > >
> > > > > We understand your concern, and you are correct that $p$ is an additional hyperparameter, but we would like to emphasize that the vast majority of new methods that the field adopts come with new hyperparameters. Additionally, please note that we do tune the relevant hyperparameters of our baselines, and perform extensive investigations into the effects of the new aforementioned hyperparameter.
> > > > >
> > > > > > for each iteration of search on $p$, one GFN network and one Q network needs to be trained, it is an additional burden compared with GFN only.
> > > > >
> > > > > We agree that training both $Q$ and GFN is a valid concern, in principle, but as we've noted here and in our paper's _Limitations_, training $Q$ and GFN does not take much more time; specifically only 5-10% more, which did not seem worth reporting precisely.
> > > > >
> > > > > This negligibly increased training time may be well worth it given the extra performance. We again would like to note that this is standard practice in ML, to pay more compute to get more performance. Even if QGFN took twice as long to train, if it means 3x more high-reward candidates, it seems to us like a very reasonable trade-off.
> > > > >
> > > > > > Comparing with modified GFN, such as local search GFN, LED-GFN, QGFN seems to be more computationally expensive by an additional Q, and the paper and the rebuttal do not provide a comparison
> > > > >
> > > > > We take note of your point that we do not report runtimes in the paper, and will be sure to include this in our revision.
> > > > >
> > > > > > Q function could help improve these modified GFN but no experiments except for FL-GFN, a relatively weaker baseline, either
> > > > >
> > > > > We ran FL-GFN as per your suggestion for testing on MIS. Given the limited time available, we used the public codebase which contains FL-GFN, and we did see improvement. This is additional evidence (we e.g. ran QGFN with SubTB and FM as shown in Figure 12 and 13) that QGFN can apply to any GFN.
> > > > >
> > > > > Unless there is a specific reason for which you believe that QGFN would not improve GFN variants such as LED-GFN, it would be beyond the scope of this paper to expect an implementation of all the now numerous GFN variants. A priori, QGFN would improve them, and we've demonstrated evidence of the mechanisms through which QGFN would achieve that. LED-GFN, for example, is not immune to the example given in Figure 1 of our paper (which is analogous to real-world dense-reward settings).
> > > > >
> > > > > We urge you to reconsider your evaluation in light of QGFN's contributions and the evidence presented in this paper.

---

> ### Comment · Reviewer_eVPz · 2024-08-14
>
> Thanks for your response. Regarding the additional $Q$, I think training $Q$ and GFN is only 5-10% more compared with standard GFN is very important to justify the inclusion of the additional $Q$. Because modified GFNs also require more training time than standard GFNs. I raise the score to 5. The reason I do not raise to the higher score is the following.
>
> > Even if QGFN took twice as long to train, if it means 3x more high-reward candidates.
>
> But the problem is that it does not have so much performance gain, especially when the lack of a stronger baseline.
>
> In the last part, I actually mean that this paper should contain more baselines like local search GFN, LED GFN, and compare QGFN directly with them (not incorporate $Q$ into them). Since they are the most up-to-date GFN variants and do not require an additional $Q$. The claim that $Q$ can improve these GFN variants is additionally from your rebuttal, not from the reviewers. Regarding these two claims, that "Q is better than GFN variants" and "Q can improve the GFN variants",  the only result is that FL+Q is slightly better than FL. Again, I understand the limited rebuttal time can be an issue, but the first claim should not require much coding to verify given these methods are all open-source.

---

### Official Review · Reviewer_Fo8k · 2024-07-02

**Soundness:** 3
**Presentation:** 4
**Contribution:** 3
**Rating:** 7
**Confidence:** 4

**Summary:**

The paper focuses on improving high-reward sample collection, i.e., exploitation, in training GFlowNets. The motivation stems from the fact that the flow may pursue states that lead to many low-reward states rather than focusing on states that lead to high-reward states. To this end, the authors propose incorporating Q-value estimation, i.e., expected future reward, and interpolating between flow and Q to make a transition. Experimentally, the proposed method shows promising performance in fragment and atom-wise molecule generation and RNA-sequence generation.

**Strengths:**

- The paper is well-written and easy to follow, with a well-developed motivation and idea.
- It seems interesting that the compositional nature of the generation model can be considered to improve the performance of GFlowNets for practical applications.
- This paper discusses various design choices for incorporating Q-value estimation, which offer different perspectives on "greediness."
- Overall, the experiments are well-done, and the reported results demonstrate the effectiveness of the proposed model.

**Weaknesses:**

No major weaknesses. See questions below.

**Questions:**

- Are there failure scenarios where QGFN shows poor performance compared to GFN? (opposite of Figure 1)
- I am curious if QGFN performs better than a policy interpolated between GFlowNets trained with low temperature and high temperature (with $p$).
- Is there an analytic form of the sampling distribution of terminal states for the proposed methods?
- It would be interesting to illustrate the sample efficiency for discovering modes in a toy example, e.g., hyper-grid.
- Line 160: "good.." => "good."

**Limitations:**

The authors have acknowledged the limitations of their approach.

---

> ### Author Rebuttal · Authors · 2024-08-07
>
> Thank you for taking the time to read our paper. We appreciate your questions and have provided detailed answers below.
>
> > Are there failure scenarios where QGFN shows poor performance compared to GFN? (opposite of Figure 1)
>
> We haven't encountered such scenarios in practice other than $p$ being set too high. In such cases, the generated samples tend to be overly similar to each other, reducing the diversity and generalization of the model.
>
> Additionally, it is possible that QGFN could show poor performance if the underlying GFN is improperly trained. If the GFN is suboptimal, the combination of $Q$ could amplify this problem.
>
> > I am curious if QGFN performs better than a policy interpolated between GFlowNets trained with low temperature and high temperature (with $p$).
>
> While we haven't performed this exact experiment, we can extrapolate from Figure 6b that this interpolation would still be less diverse than QGFN, stemming from the loss in diversity in low-temperature (high $\beta$) GFN.
>
> > Is there an analytic form of the sampling distribution of terminal states for the proposed methods?
>
> We haven't been able to derive any interesting results in this respect. In the limits of $p$ one simply retrieves either the GFN behavior or the greedy $Q$ policy behavior. Note that in some restricted settings (think of the bandit case for example), sampling from QGFN is provably at least as good as sampling from GFN. We include such an analysis as well as intuition-building for the general case in the global response.
>
> > It would be interesting to illustrate the sample efficiency for discovering modes in a toy example, e.g., hyper-grid.
>
> While the hyper-grid example has been used in many GFlowNet papers, we do not believe it to be particularly interesting other than as a sanity check of convergence; i.e. it should not be used to rank algorithms. The main reason is that the setting is too simple to get much generalization out of the DNN approximating $P_F$ and/or $F$ (c.f. [1]). We can nonetheless add such an experiment to the appendix as an additional sanity check.
>
> [1] Investigating Generalization Behaviours of Generative Flow Networks, Lazar Atanackovic, Emmanuel Bengio, 2024

---

> > ### Comment · Reviewer_Fo8k · 2024-08-08
> >
> > Thank you for the detailed clarifications. I have no further comment.

---

### Official Review · Reviewer_bgQq · 2024-07-15

**Soundness:** 2
**Presentation:** 2
**Contribution:** 2
**Rating:** 4
**Confidence:** 5

**Summary:**

The paper proposes jointly learning a $Q$ function and a policy network $P_F$ to improve the search for high-valued states when training GFlowNets. To achieve this, the authors develop three sampling strategies for composing $Q$ and $P_F$, $p$-greedy, $p$-of-max, and $p$-quantile, and show that the resulting algorithm, termed QGFN, often leads to faster discovery of modes relative to GFlowNet baselines.

In spite of QGFN’s notable performance, I believe the paper would greatly benefit from a more clearly described solution and from the inclusion of stronger, optimization-focused baselines. I will be happy to increase my score if the enumerated weaknesses and questions below are properly addressed during the rebuttal period.

**Strengths:**

1. **Well-described problem**. Section 3 highlights the issues of an exclusively GFlowNet-oriented approach to the search of high-valued states with clarity, namely, the large probability mass associated with low-reward states.

2. **Intuitively sound method with strong empirical results**. When the $Q$ function is accurately learned, QGFN — which interpolates between a GFlowNet (p = 0) and a DQN (p = 1) — should yield samples with higher rewards for larger $p$. This behavior is experimentally confirmed. However, the distribution from which QGFN samples, even when $Q$ is perfectly estimated, is mostly unclear. See weakness 4 below.

3. **Extensive experimental campaign**. (however, important baselines are missing) Experiments include four commonly used benchmark tasks for GFlowNets, and QGFN often outperforms alternative methods by a large margin.

**Weaknesses:**

1. **Missingness of important baselines**. FL-GFlowNets (Pan et al., 2023) and LED-GFlowNets (Jang et al., 2024) exhibited strong performance for the optimization-focused tasks of molecule and set generation. Notably, contrarily to the greediness $p$ of QGFN, these methods do not require tuning of a hyperparameter that may unpredictably modify the sampling distribution. Authors should include FL-GFlowNets and LED-GFlowNets in Figures 2, 3, and 4. Have the authors considered increasing the sharpness of the distribution by modifying the temperature in LSL-GFN?

2. **Training of the $Q$-network is insufficiently described**. Lines 163-175 are hard to follow and Algorithm 1 in Section B does not provide sufficient details for understanding the training of QGFN. If I understand correctly, the adoption of multi-step returns with the horizon size set as the maximum trajectory length implies that the each $R(s_{t})$ in Equation (2) corresponds to the expected reward among $s_{t}$’s children under the chosen sampling policy. In any case, an unambiguous equation representing the learning objective for $Q$ should be included in Section 4.

3. **Difference between each sampling strategy is unclear**. In practical applications, a sampling strategy would have to be chosen, as selecting among the proposed approaches can be excessively time-consuming — even without retraining the model. However, $p$-greedy, $p$-of-max, and $p$-quantile seem to perform relatively different when the benchmark task is modified. A general approach for choosing a sampling strategy should, then, be proposed by the authors. Otherwise, the method is hardly usable in real applications.

4. **QGFNs lack theoretical guarantees**. Important questions are left unanswered. What if $Q$ is not properly trained? How do we know that $Q$ is inaccurately learned? What is the sampling for a given $p$ and sampling strategy? To improve QGFN’s reliability, a theoretical analysis should be included. Does a sufficiently trained QGFN finds, in average, more rewarding than standard GFlowNets?

5. **Rationale for choosing $p$ is not well-defined**. Experiments are filled with cryptically chosen numbers. In Table 1, the $p$ value for $p$-of-max and $p$-quantile are respectively set to $0.9858$ and $0.93$; in Figure 3, a “mode” is defined as a molecule with Tanimoto similarity smaller than $0.70$ and a reward larger than $1.10$. What is the rationale behind such numbers? How can a practitioner choose $p$? Also, if the choice of $p$ relied on a grid search over $N$ values in $[0, 1]$, a fair comparison with GFN-TB and GFN-SubTB in Table 1 should allow these methods to sample $N$x times more trajectories than QGFN. However, it is unclear whether this observation was considered for the experiments defining Table 1.

6. [Minor] Typos. Algorithm 1 refers to $\mathcal{L}\_{flow}$ and $\mathcal{L}\_{huber}$, which are undefined. I assume $\mathcal{L}\_{flow}$ is $\mathcal{L}\_{TB}$, but I could not find a definition for $\mathcal{L}\_{huber}$.

**Questions:**

* Why does QGFN provide more diverse samples than, e.g., GFN-TB in Table 1 even when $p$ is set as high as $p = 0.9858$?
Do we need to train a GFN? Given the large values of $p$ (above 0.9), it seems that the GFN is not very helpful. I wonder whether an untrained $P_F$ would produce similar results.

* What is $k$ in line 230?

* Figures 9 and 11 are quite difficult to grasp. Please consider visually encoding the greediness of the method as the size or opacity of points in a scatter plot.

* Also, the convergence speed of GFlowNets via TB minimization is known to be dramatically affected by the learning rate of $\log Z$, which the authors set to $10^{-3}$. I wonder whether the relative results would remain the same if such learning rate was increased to $10^{-2}$ or $10^{-1}$.

**Limitations:**

Authors claim that the computational overhead induced by learning a Q network and the sensitivity of sampling to the accuracy of $Q$ are the main limitations of their work.

---

> ### Author Rebuttal · Authors · 2024-08-07
>
> Thank you for your feedback. We appreciate the time and effort you put into reviewing our work.
>
> Our work aims to provide an empirical analysis of the method we introduce, a novel idea that can be applied to any GFN. While we understand the reviewer's emphasis on theoretical guarantees, it is worth noting that many significant contributions in the field do not include such guarantees. We believe that our approach, with thorough empirical analysis, is a strong contribution to the field and aligns with the standards of many accepted papers in major ML conferences.
>
> Overall, we hope to have addressed the reviewer's concerns in our response, and hope they would reconsider their score. We are committed to improving the presentation of the paper, particularly to dispel any sense of choices and (hyper)parameters being arbitrary.
>
> > baselines, temperature in LSL-GFN
>
> We compared against six baselines typically used in the GFN literature, covering characteristics like greediness and diversity. Further, the QGFN idea can be applied to any GFN, including FL-QGFNs and LED-QGFNs. In this spirit, we trained FL-QGFNs for Table A of Reviewer eVPz's rebuttal. Yes, if a base GFN algorithm is already optimal, QGFN may not improve it, but this is unlikely to be the case in general. Our contribution is to show that mixtures of action values and GFN policies have desirable behaviors; we did not intend to show that a specific QGFN implementation is SoTA.
>
> For LSL-GFN, we tested the effects of increasing sharpness during inference using a range of $\beta$ values (training $\beta\sim U[1,256]$), and chose $\beta = 78$, with rewards closest to SubTB for comparison (larger $\beta$ greatly increased similarity). We will include these results in our revision in Table 1 (see attached pdf).
>
> > Training of the $Q$-network
>
> We will include the exact objective of $Q$ in $\S$ 4 and Alg. 1. The reviewer is correct that in the max horizon case $Q_\theta(s,a)$ converges to the expected reward. This is the crux of lines 163-175, where we attempt to convey this unexpected reliance on max horizon returns.
>
> > Difference between strategies
>
> Section 5.1 highlights that the best strategy depends on the environment, particularly the number of relevant actions. Although not foolproof, this is a reasonable starting point. That QGFN is adjustable post-hoc is a strength of the method which can be used here.
>
> > QGFNs lack theoretical guarantees
>
> - the distribution from which a perfect QGFN samples is hard to define theoretically. We would be happy to find such a result, but we would ask the reviewer to not hold us to impossible standards. For example, Schulman et al. did not know what distribution PPO sampled from, but produced an empirically well understood method. It took two years for proper analysis using two-time-scale methods to provide us with strong theory
> - In specific cases one can show that the expected reward of QGFN is at least as much as GFN's. We include analysis for the bandit case and intuition-building for the general case in the global response. We would like to emphasize that we've performed extensive empirical work beyond measuring performance to show that our method indeed works according to its hypothesized mechanism
> - _What if $Q$ is not properly trained?_ Early in training we observe that QGFNs tend to outperform baselines. At that stage it is unlikely that $Q$ is precise.
> - _How do we know if $Q$ is accurate?_ This can be measured, e.g. in Fig 7 showing an imperfect $Q$. Note that this $Q$ is still improving the sampling of high-reward diverse objects
>
> On $p$ and other values:
>
> - as explained in $\S5.1$ 244-288 and $\S6$ 280-286, and shown in Figure 6, the $p$s of Table 1 are chosen based on Figure 6.
> - _Why this Tanimoto threshold?_ 0.7 has been used in every GFlowNet paper with this task, we simply reuse this choice from the literature.
> - _How can one choose $p$?_ We are unsure what the reviewer means here. There are standard ways to choose (hyper)parameters, but we show in Fig 6 and 9 that e.g. sampling from the model with different $p$s and measuring the diversity-reward trade-off works
> - _baselines should be allowed to sample $N$x times._ This is incorrect, because we are reporting averages. As a thought experiment, if we sampled infinitely many samples for them, the _average_ reward and similarity would converge to some values. Table 1 uses 1000 samples.
>     - A better test would be to log the time of training QGFN + tuning $p$. The cost of training $Q$ is small (thank parallelism), and taking ~1000 samples with different $p$s takes a minute. That QGFN is adjustable without any retraining is a strength of the method
>
> Questions:
> - _Why is QGFN [with $p=0.9858$] more diverse than GFN-TB?_ Note that this is for **$p$-of-max QGFN**: if many actions have a $Q$ value that is at least 98.58% of the max $Q$, then they can all be sampled by $P_F$, retaining the diversity of $P_F$, but sampling more high-reward objects. Note QGFN isn't strictly more diverse than GFN, but for $n$ samples its _number of modes_ will be higher because less time is spent sampling low-rewards
> - _with large $p$, is QGFN helpful?_ As above, a good $P_F$ is still needed, and a $p$ close to but less than 1 does **not** entail the greedy policy, except for the $p$-greedy mixture (where $p\approx 0.5$ are best; see Fig 9 & l287-292)
> - _Would an untrained $P_F$ be similar?_ This is an interesting question! We compared a trained $P_F$ to an untrained $P_F$ (trained $Q$ for both). The latter has an ok performance for high $p$, but is still far from what QGFN can reach (see attached pdf Fig. 1)
> - $k$ in line 230 refers to the bit width of actions that we later use (lines 259-261) as per Malkin et al. We will clarify this
> - Style of Fig. 9&11, we were uncertain about this, and use a revised style in Fig. 1 of the attached pdf. We would welcome feedback
> - For us large learning rates for $Z$ were not helpful, and too often caused runs to diverge

---

> > ### Author Response · Authors · 2024-08-10
> > **Clarification on $p$ and other values**
> >
> > We apologize for a mistake in our rebuttal, inference time $p$ values are chosen based on Figure 9, not Figure 6. Figure 6's left two figures use these inference-time $p$ values, while Figure 6's 3rd subplot show the effect of different (fixed) $p$ values used for training.

---

> > > ### Comment · Reviewer_bgQq · 2024-08-14
> > >
> > > Thank you for the responses and additional experiments. Also, sorry for the delayed response.
> > >
> > > > 0.7 has been used
> > >
> > > Thank you for the clarification. Some works (e.g., [1, 2]) do not seem to explicitly report this when training a GFlowNet on the QM9 environment. A reference would be helpful here.
> > >
> > > > This is incorrect, because we are reporting averages. As a thought experiment, if we sampled infinitely many samples for them, the average reward and similarity would converge to some values.
> > >
> > > Indeed, for infinitely many samples, all reported metrics would perfectly reflect the trained model’s underlying distribution. Your results, however, only cover the finite case representing a small fraction of the state space. For the average reward in Table 1, the difference may be harmless; for the number of modes in Figure 4, the difference is critical.
> > >
> > > > We are unsure what the reviewer means
> > >
> > > My question is related to the comment above. If I implement, e.g., a grid search with $N$ candidates to find the best $p$ for a given task, and in each run I sample $T$ trajectories from the trained model, then I would have sampled $NT$ trajectories in total, right?
> > >
> > > When my interest lies in finding high-valued and diverse samples, will this approach be more effective than directly sampling $NT$ trajectories from a standardly trained GFlowNet? The results in the paper do not seem to provide evidence for this. In other words, the question I ask is: if $p$ cannot be generally chosen prior to inference, is QGFN beneficial given the extra computation cost of hyperparameter selection?
> > >
> > > > Style of Fig. 9&11
> > >
> > > Thank you. In my view, the figure attached to the rebuttal PDF is significantly better than the ones reported in the main text.
> > >
> > > While I understand that the discussion period is near the end and that there is no time remaining for further discussion, I believe that crucial aspects of the author’s work, e.g., fair comparison to baselines and clarity of presentation, were not properly addressed.
> > >
> > > [1] Learning to Scale Logits for Temperature-Conditional GFlowNets. Kim et al. ICML 2024.
> > >
> > > [2] Towards Understanding and Improving GFlowNet Training. Shen et al. ICML 2023.

---

### Author Rebuttal · Authors · 2024-08-07

We appreciate all feedback from the reviewers. To provide more analysis on QGFN, we have conducted an analysis using a bandit example. We also use some derivations to illustrate the general case. We hope these examples illustrate the behavior of QGFN. We will include this analysis in our revised manuscript.

### Analysing $p$-greedy

Consider the bandit setting where trajectories are 1 step and just consist in choosing a terminal state. Let $p_G(s) = R(s)/Z$. Let $0<p<1$, then with $\mu(s'|s) = (1-p)P_F(s'|s) + p\mathbb{I}[s'=\arg\max Q(s,s')]$, assuming there is only a single argmax $s^*$, then $p_\mu(s) = (1-p)R(s)/Z + p\mathbb{I}[s=\arg\max R(s)]$. This means that for every non-argmax state, $p_\mu(s) = (1-p) p_G(s) < p_G(s)$. We get that $\mathbb{E}_\mu[R] > \mathbb{E}_G[R]$:

$$\\begin{align}
\\mathbb{E}_ \\mu[R] - \\mathbb{E}_ G[R]=&\\sum_ s p_ \\mu(s) R(s) - \\sum_s p_G(s) R(s)\\\\
=& (p + (1-p)R(s^*)/Z - R(s^*)/Z) R(s^*) + \\sum_{s\\neq s^*} (1-p)R(s)^2/Z-R(s)^2/Z\\\\
=& pR(s^*) - pR(s^*)^2/Z  + \\sum_{s\\neq s^*} (-p)R(s)^2/Z\\\\
=& p/Z \\left(R(s^*)Z - R(s^*)^2  - \\sum_{s\\neq s^*} R(s)^2\\right),\\;\\;\\;Z=\\sum_s R(s)\\\\
=& p/Z \\left(R(s^*)[\\sum_s R(s)] - R(s^*)^2  - \\sum_{s\\neq s^*} R(s)^2\\right)\\\\
=& p/Z \\left(R(s^*)[\\sum_s R(s)] - \\sum_{s} R(s)^2\\right)\\\\
=& p/Z \\left(\\sum_{s} R(s^*)R(s) - R(s)^2\\right)\\\\
\\end{align}$$

since $R(s^*) > R(s)$ and both are positive then $R(s^*)R(s) > R(s)^2$ thus the last sum is positive. All other terms are positive, therefore $\mathbb{E}_\mu[R] - \mathbb{E}_G[R] > 0$.

In the more general case, we are not aware of a satisfying closed form, but consider the following exercise.

Let $m(s,s') = \mathbb{I}[s'=\arg\max Q(s,s')]$. Let $F'$ be the "QGFN flow" which we'll decompose as $F'=F_G + F_Q$ where we think of $F_G$ and $F_Q$ as the GFN and Q-greedy contributions to the flows. Then:

$$\begin{align}
F'(s) &= \\sum_{z\\in\\mathrm{Par}(s)} F'(z)((1-p)P_F(s|z) + pm(z,s))\\\\
&=\\sum_z F'(z)(1-p)P_F(s|z) + \\sum_z F'(z) pm(z,s)\\\\
&=\\sum_z F_G(z)(1-p)P_F(s|z) + F_Q(z)(1-p)P_F(s|z)+ \\sum_z F'(z) pm(z,s)\\\\
&=(1-p)F_G(s) + \\sum_z F_Q(z)\\mu(z|s)+ F_G(z) pm(z,s)\\\\
\end{align}$$

Recall that $p(s)\propto F(s)$. Intuitively then, the probability of being in a state is reduced by a factor $(1-p)$, but possibly increased by this extra flow that has two origins. First, flow $F_Q$ carried over by $\mu$, and second, new flow being "stolen" from $F_G$ from parents when $m(z, s)=1$, i.e. when $s$ is the argmax child.

This suggests that flow (probability mass) in a $p$-greedy QGFN is smoothly redirected towards states with locally highest reward from ancestors of such states. Conversely, states which have many ancestors for which they are not the highest rewarding descendant will have their probability diminished.

---

### Decision · Program_Chairs · 2024-09-25

**Decision:**

Accept (poster)

**Comment:**

This paper proposes QGFN, which jointly trains G-flownets and Q-functions and combines their predictions with the goal of sampling diverse and high-reward objects.

Several important points were raised by the reviewers, which were partially addressed during the rebuttal phase:

- Training details for the Q-network are not sufficiently described (reviewer ``bgQq``).

- Additional complex environment experiments are needed on MIA [1].

Overall, my recommendation is to accept the paper. I suggest the authors incorporate the feedback in the final submission.

[1] Zhang, D., Dai, H., Malkin, N., Courville, A., Bengio, Y., & Pan, L. (2023). Let the Flows Tell: Solving Graph Combinatorial Optimization Problems with G-Flownets. arXiv preprint arXiv:2305.17010.